# A diagram for evaluating multiple aspects of model performance in simulating vector fields

Zhongfeng Xu[1], Zhaolu Hou[2,3,4], Ying Han[1], Weidong Guo[2]

[1] RCE-TEA, Institute of Atmospheric Physics, Chinese Academy of Sciences, Beijing 100029, China
[2] School of Atmospheric Sciences, Nanjing University, Nanjing 210093, China
[3] LASG, Institute of Atmospheric Physics, Chinese Academy of Sciences, Beijing 100029, China
[4] University of Chinese Academy of Sciences, Beijing 100049, China

*Correspondence to*: Zhongfeng Xu (xuzhf@tea.ac.cn)

**Abstract.** Vector quantities, e.g., vector winds, play an extremely important role in climate systems. The energy and water exchanges between different regions are strongly dominated by wind, which in turn shapes the regional climate. Thus, how well climate models can simulate vector fields directly affects model performance in reproducing the nature of a regional climate. This paper devises a new diagram, termed the vector field evaluation (VFE) diagram, which is a generalized Taylor diagram and able to provide a concise evaluation of model performance in simulating vector fields. The diagram can measure how well two vector fields match each other in terms of three statistical variables, i.e., the vector similarity coefficient, root-mean-square length (RMSL), and root-mean-square vector difference (RMSVD). Similar to the Taylor diagram, the VFE diagram is especially useful for evaluating climate models. The pattern similarity of two vector fields is measured by a vector similarity coefficient (VSC) that is defined by the arithmetic mean of the inner product of normalized vector pairs. Examples are provided, showing that VSC can identify how close one vector field resembles another. Note that VSC can only describe the pattern similarity, and it does not reflect the systematic difference in the mean vector length between two vector fields. To measure the vector length, RMSL is included in the diagram. The third variable, RMSVD, is used to identify the magnitude of the overall difference between two vector fields. Examples show that the new diagram can clearly illustrate the extent to which the overall RMSVD is attributed to the systematic difference in RMSL and how much is due to the poor pattern similarity.

Keywords: vector field similarity, root-mean-square vector difference, root-mean-square length, model evaluation

## 1 Introduction

Vector quantities play a very important role in climate systems. It is well known that atmospheric circulation transfers mass, energy, and water vapor between different parts of the world, which is an extremely crucial factor to shaping regional climates. The monsoon climate is a typical example of one that is strongly dominated by atmospheric circulation. A strong Asian summer monsoon circulation usually brings more precipitation and vice versa. Therefore, the simulated precipitation is strongly determined by how well climate models can simulate atmospheric circulation (Twardosz et al. 2011; Sperber et al., 2013; Zhou et al., 2016; Wei et al., 2016). Ocean surface wind stress is another important vector quantity that reflects the momentum flux between the ocean and atmosphere, serving as one of the major factors for oceanic circulation (Lee et al., 2012). The wind stress errors can cause large uncertainties in ocean circulation in the subtropical and subpolar regions (Chaudhuri et al., 2013). Thus, the evaluation of vector fields, e.g., vector winds and wind stress, would also help in understanding the causes of model errors.

The Taylor diagram (Taylor 2001) is very useful in evaluating climate models, and it has been widely used in model inter-comparison and evaluation studies over the past several years (e.g., Hellström and Chen, 2003; Martin et al., 2011; Giorgi and Gutowski, 2015; Jiang et al., 2015; Katragkou et al. 2015). However, the Taylor diagram was constructed for evaluating scalar quantities, such as temperature and precipitation. The statistical variables used in Taylor diagram, i.e., the Pearson correlation coefficient, standard deviation, and centered root-mean-square error (RMSE), do not apply to vector quantities. No such diagram is yet available for evaluating vector quantities such as vector winds, wind stress, temperature gradients, and vorticity. Previous studies have usually assessed model performance in reproducing a vector field by evaluating its x- and y-component with the Taylor diagram (e.g., Martin et al. 2011; Chaudhuri et al., 2013). Although such an evaluation can also help to examine the modeled vector field, it suffers from some deficiencies as follows: (1) a good correlation in the x- and y-component of the vector between the model and observation may not necessarily indicate that the modeled vector field resembles the observed one. For example, assuming we have two identical 2-dimensional vector fields $\vec{A}$ and $\vec{B}$, their correlation coefficients are 1 for both the x- and y-component. If the x-component of vector field $\vec{A}$ adds a constant value, the correlation coefficients for both the x- and y-component do not change, but the direction and length of vector $\vec{A}$ change, which suggests that the pattern of two vector fields are no longer identical. Thus, computing the correlation coefficients for the x- and y-component of a vector field is not well suited for examining the pattern similarity of two vector fields. (2) It is hard to determine the improvement of model performance. For example, should one conclude that the model performance is improved if the RMSE (or correlation coefficient) is reduced for the y-component but increased for the x-component of a vector field? Given these reasons and the importance of vector quantities in a climate system, we have developed a new diagram, termed the vector field evaluation (VFE) diagram, to measure multiple aspects of model performance in simulating vector fields.

To construct the VFE diagram, one crucial issue is quantifying the pattern similarity of two vector fields. Over the past several decades, many vector correlation coefficients have been developed by different approaches. For example, some vector correlation coefficients are constructed by combining Pearson's correlation coefficient of the x- and y-component of the vector (Charles, 1959; Lamberth, 1966). Some vector correlation coefficients are devised based on orthogonal decomposition (Stephens, 1979; Jupp and Mardia, 1980; Crosby et al., 1993) or the regression relationship of two vector fields (Ellison, 1954; Kundu, 1976; Hanson et al., 1992). These vector correlation coefficients usually do not change when one vector field is uniformly rotated or reflected to a certain angle. This is a reasonable and necessary property for the vector correlation coefficient when one detects the relationship of two vector fields. However, in terms of model evaluation, we expect the simulated vectors to resemble the observed ones in both direction and length with no rotation permitted. Thus, previous vector correlation coefficients are not well suited for the purpose of climate model inter-comparisons and evaluation.

To measure how well the patterns of two vector fields resemble each other, a vector similarity coefficient (VSC) is introduced in section 2 and interpreted in section 3. Section 4 constructs the VFE diagram with three statistical variables to evaluate multiple aspects of simulated vector fields. Section 5 illustrates the use of the diagram in evaluating climate model performance. A discussion and conclusion are provided in section 6.

## 2 Definition of vector similarity coefficient

Consider two vector fields $\vec{A}$ and $\vec{B}$ (Figure 1a). Without loss of generality, vector field $\vec{A}$ and $\vec{B}$ can be written as a pair of vector sequences:

$\vec{A}_i = (x_{ai}, y_{ai}); \ i = 1, 2, \ldots, N$

$\vec{B}_i = (x_{bi}, y_{bi}); \ i = 1, 2, \ldots, N$

Each vector sequence is composed of N vectors. To measure the similarity between vector fields $\vec{A}$ and $\vec{B}$, a vector similarity coefficient (VSC) should be able to recognize how much and to what degree the vectors are in the same direction and the vector lengths are proportional to each other. Thus VSC is defined as follows:

$$R_v = \frac{\sum_{i=1}^{N} \vec{A}_i \cdot \vec{B}_i}{\sqrt{\sum_{i=1}^{N} |\vec{A}_i|^2} \sqrt{\sum_{i=1}^{N} |\vec{B}_i|^2}} \qquad (1)$$

where || represents the length of a vector. · represents the inner product.

We define a normalized vector as follows:

$$\vec{A}_i^* = \frac{\vec{A}_i}{\sqrt{\frac{1}{N}\sum_{i=1}^{N}|\vec{A}_i|^2}} = \frac{\vec{A}_i}{L_A} \quad (2)$$

and

$$\vec{B}_i^* = \frac{\vec{B}_i}{\sqrt{\frac{1}{N}\sum_{i=1}^{N}|\vec{B}_i|^2}} = \frac{\vec{B}_i}{L_B} \quad (3)$$

respectively, where

$$L_A = \sqrt{\frac{1}{N}\sum_{i=1}^{N}|\vec{A}_i|^2} \quad (4)$$

and

$$L_B = \sqrt{\frac{1}{N}\sum_{i=1}^{N}|\vec{B}_i|^2} \quad (5)$$

are the quadratic mean of the length or root-mean-square length (RMSL) of a vector field which measures the mean and variance of vector lengths (Equation A1). Based on (2) and (3), we have

$$\sum_{i=1}^{N}|\vec{A}_i^*|^2 = \sum_{i=1}^{N}|\vec{B}_i^*|^2 = N \quad (6)$$

Clearly, the normalization of a vector field only scales the vector lengths without changing their directions (Fig. 1b).

With the aid of (2) and (3), equation (1) can be rewritten as

$$
\begin{aligned}
R_v &= \frac{1}{N}\sum_{i=1}^{N}\vec{A}_i^* \cdot \vec{B}_i^* \\
&= \frac{1}{N}\sum_{i=1}^{N}|\vec{A}_i^*||\vec{B}_i^*|\cos\alpha_i \\
&= \frac{1}{N}\sum_{i=1}^{N}\frac{|\vec{A}_i^*|^2 + |\vec{B}_i^*|^2 - |\vec{C}_i^*|^2}{2} \quad (7) \\
&= 1 - \frac{1}{2N}\sum_{i=1}^{N}|\vec{C}_i^*|^2 \\
&= 1 - \frac{1}{2}\text{MSDNV}
\end{aligned}
$$

where $\vec{C}_i^*$ is the difference between the normalized $\vec{A}$ and $\vec{B}$ (Fig. 1b). MSDNV is mean-square difference of the normalized vectors (Shukla and Saha 1974, with minor modification) between two normalized vector sequences:

$$\text{MSDNV} = \frac{1}{N}\sum_{i=1}^{N}\left|\vec{A}_i^* - \vec{B}_i^*\right|^2 = \frac{1}{N}\sum_{i=1}^{N}\left|\vec{C}_i^*\right|^2 \qquad (8)$$

Given the triangle inequality, $0 \leq \left|\vec{C}_i^*\right| \leq \left|\vec{A}_i^*\right| + \left|\vec{B}_i^*\right|$, we have

$$0 \leq \left|\vec{C}_i^*\right|^2 \leq \left(\left|\vec{A}_i^*\right| + \left|\vec{B}_i^*\right|\right)^2 \leq 2\left|\vec{A}_i^*\right|^2 + 2\left|\vec{B}_i^*\right|^2 \qquad (9)$$

With the aid of (6), (7), (8) and (9) we obtain

$$0 \leq \text{MSDNV} \leq 4, \text{ and } -1 \leq R_v \leq 1$$

$R_v$ reaches its maximum value of 1 when MSDNV = 0, i.e., $\vec{A}_i^* = \vec{B}_i^*$ for all i $(1 \leq i \leq N)$. $R_v$ reaches its minimum value of $-1$ when MSDNV = 4, i.e., $\vec{A}_i^* = -\vec{B}_i^*$ for all i $(1 \leq i \leq N)$. Thus, the vector similarity coefficient, $R_v$, always takes values in the intervals $[-1, 1]$ and is determined by MSDNV, namely $\frac{1}{N}\sum_{i=1}^{N}\left|\vec{C}_i^*\right|^2$. Clearly, $\left|\vec{C}_i^*\right|$ is determined by the differences in both vector lengths and angles between $\vec{A}_i^*$ and $\vec{B}_i^*$ (Fig. 1b). A smaller $\left|\vec{C}_i^*\right|$ suggests that $\vec{A}_i^*$ is closer to $\vec{B}_i^*$ and vice versa. To

10 better understand $R_v$, some special cases are discussed as follows.

For all i $(1 \leq i \leq N)$:

If $\vec{A}_i^* = \vec{B}_i^*$, then $\left|\vec{C}_i^*\right| = 0$. We obtain $R_v = 1$ when each pair of normalized vectors is exactly the same length and direction.

If $\vec{A}_i^* = -\vec{B}_i^*$, then $\left|\vec{A}_i^*\right| = \left|\vec{B}_i^*\right| = \left|\vec{C}_i^*\right|/2$. We obtain $R_v = -1$ when each pair of normalized vectors is exactly the same length but opposite direction.

If $\vec{A}_i^* \perp \vec{B}_i^*$, then $\left|\vec{A}_i^*\right|^2 + \left|\vec{B}_i^*\right|^2 = \left|\vec{C}_i^*\right|^2$. We obtain $R_v = 0$ when each pair of normalized vectors is orthogonal to each other.

If $\left|\vec{C}_i^*\right|^2 < \left|\vec{A}_i^*\right|^2 + \left|\vec{B}_i^*\right|^2$, we obtain $0 < R_v < 1$ when the angles between $\vec{A}_i^*$ and $\vec{B}_i^*$ are acute angles.

If $\left|\vec{C}_i^*\right|^2 > \left|\vec{A}_i^*\right|^2 + \left|\vec{B}_i^*\right|^2$, we obtain $-1 < R_v < 0$ when the angles between $\vec{A}_i^*$ and $\vec{B}_i^*$ are obtuse angles.

Thus, a positive (negative) $R_v$ indicates that the angles between $\vec{A}_i^*$ and $\vec{B}_i^*$ are generally smaller (larger) than 90°, which suggests that the patterns between $\vec{A}_i^*$ and $\vec{B}_i^*$ are similar (opposite) to each other. A greater $R_v$ indicates a higher similarity

between two vector fields. Based on equations (2), (3), and (7), $R_v$ does not change when $\vec{A}$ or $\vec{B}$ is multiplied by a positive constant, which is analogous to the property of Pearson's correlation coefficient. Thus, $R_v$ can measure the pattern similarity of two vector fields but cannot determine whether two vector fields have the same amplitude in terms of the mean length of vectors. However, we can use the scalar variable RMSL (equation 4 and 5) to measure the mean length of a vector field.

## 3 Interpreting VSC

In this section, we present three cases to explain why VSC can reasonably measure the pattern similarity of two vector fields. To facilitate the interpretion, we define the mean difference of angles (MDA) between paired vectors as follows:

$$\text{MDA} = \bar{\alpha} = \frac{1}{N}\sum_{i=1}^{N}\alpha_i = \frac{1}{N}\sum_{i=1}^{N}\text{acos}\left(\frac{\vec{A}_i \cdot \vec{B}_i}{\left|\vec{A}_i\right|\left|\vec{B}_i\right|}\right) \qquad (10)$$

Where the vector fields $\vec{A}$ and $\vec{B}$ are the same as those in the equation (1). $\alpha_i$ is the included angle between paired vectors. MDA takes values in intervals $[0, \pi]$ and measures how close the corresponding vector directions of two vector fields are to each other. A mean square difference (MSD) of normalized vector lengths is defined as follows:

$$\begin{aligned} \text{MSD} &= \frac{1}{N} \sum_{i=1}^{N} \left( |\vec{A}_i^*| - |\vec{B}_i^*| \right)^2 \\ &= \frac{1}{N} \sum_{i=1}^{N} \left( |\vec{A}_i^*|^2 + |\vec{B}_i^*|^2 - 2|\vec{A}_i^*||\vec{B}_i^*| \right) \qquad\qquad (11) \\ &= 2 - \frac{2}{N} \sum_{i=1}^{N} |\vec{A}_i^*||\vec{B}_i^*| \end{aligned}$$

5  Given equation (6) and the Cauchy–Schwarz inequality:

$$\left( \sum_{i=1}^{N} |\vec{A}_i^*||\vec{B}_i^*| \right)^2 \leq \sum_{i=1}^{N} |\vec{A}_i^*|^2 \sum_{i=1}^{N} |\vec{B}_i^*|^2$$

we find that MSD takes on values in intervals $[0, 2]$.

For all i $(1 \leq i \leq N)$, if $|\vec{A}_i^*| = |\vec{B}_i^*|$, we have MSD = 0,

For all i $(1 \leq i \leq N)$, if $|\vec{A}_i^*||\vec{B}_i^*| = 0$, we have MSD = 2.

MSD measures how close the paired vector lengths of two normalized vector fields are to each other. Based on the definition

10  of $R_v$ (equation 7), the VSC is determined by both differences in vector lengths and angles between two groups of vectors. To interpret the nature of VSC, we will discuss how VSC will change with MSD and MDA in section 3.1 and 3.2, respectively.

**3.1 Interpreting VSC based on its equation**

VSC can be written as follows:

$$\begin{aligned} R_v &= \frac{1}{N} \sum_{i=1}^{N} \vec{A}_i^* \cdot \vec{B}_i^* \\ &= \frac{1}{N} \sum_{i=1}^{N} |\vec{A}_i^*||\vec{B}_i^*| \cos\alpha_i \\ &= \frac{1}{N} \sum_{i=1}^{N} \left[ \frac{\left( |\vec{A}_i^*|^2 + |\vec{B}_i^*|^2 \right) - \left( |\vec{A}_i^*| - |\vec{B}_i^*| \right)^2}{2} \right] \cos\alpha_i \end{aligned}$$

To examine the relationship of VSC with MSD, we assume each corresponding angle between paired vectors $\alpha_i = \alpha = $ const ($i = [1, N]$). With the support of (6) and (11) we obtain

$$R_v = \left[ 1 - \frac{1}{2N} \sum_{i=1}^{N} \left( |\vec{A}_i^*| - |\vec{B}_i^*| \right)^2 \right] \cos\alpha$$

$$= \left[ 1 - \frac{MSD}{2} \right] \cos\alpha$$

(12)

Thus, $R_v$ varies between 0 and $\cos\alpha$ due to the difference in the normalized vector length when $\alpha$ is a constant angle. $R_v$ equals 0 when $\alpha$ equals 90° regardless of the value of MSD. MSD plays an increasingly important role in determining $R_v$ when $\alpha$ approaches 0 or 180°. $R_v$ is inversely proportional to MSD, which suggests that two vector fields show a higher similarity when their corresponding normalized vector lengths are closer to each other, and vice versa. On the other hand, $R_v$ is proportional to $\cos\alpha$, suggesting a higher VSC when the directions of paired vector are closer to each other. This indicates that VSC can reasonably describe how close the normalized vector fields are by taking both vector lengths and directions into consideration simultaneously (Equation 12).

## 3.2 Interpreting VSC based on random generated samples

In previous section, the interpretation of VSC is based on the assumption that the paired vectors have a constant included angle. In this section, we will examine how VSC is affected by the difference of included angles in a more general case. Firstly, we construct a reference vector sequence, $\vec{A}$, comprising 30 vectors, i.e., $i = [1,30]$. The lengths of 30 vectors follow a normal distribution, and the arguments of 30 vectors follow uniform distribution between 0 and 360°. Secondly, we produced a new vector sequence $\vec{B}$ by rotating each individual vector of $\vec{A}$ by a certain angle randomly between 0° and 180° without changes in vector lengths. Such a random generation of $\vec{B}$ was repeated $1 \times 10^6$ times to produce sufficient random samples of vector sequences. The vector similarity coefficients $R_v$ are computed between $\vec{A}$ and the $1 \times 10^6$ sets of randomly produced vector sequences, respectively. As shown in Figure 2, $R_v$ generally shows a negative relationship with MDA, i.e., a smaller MDA generally corresponds to a larger $R_v$, and vice versa. A smaller MDA indicates smaller differences in the directions of paired vectors and hence a higher similarity between the vector fields $\vec{A}$ and $\vec{B}$, suggesting that VSC can reasonably describe how close the vector directions between two vector fields are. Meanwhile, it is also noted that $R_v$ varies within a large range for the same MDA. For example, when MDA equals 90°, $R_v$ can vary from approximately -0.5 to 0.5 depending on the relationship between the paired vector lengths and the corresponding included angles (Fig. 2). A positive (negative) $R_v$ is observed when the 30 vector lengths and included angles are negatively (positively) correlated. This means that the patterns of two vector fields are closer (opposite) to each other when the included angles between the long vectors are small (large). Specifically, the rotation of shorter vector may not undermine $R_V$ too much as long as the longer vectors remain unchanged. In contrast, $R_V$ would be strongly undermined with the rotation of longer vectors. Simply put, the longer vectors generally play a more important role than the shorter vectors in determining $R_v$.

### 3.3 Application of VSC to 850-hPa vector winds

In this section, we compute the $R_v$ of the climatological mean 850-hPa vector winds in January with that in each month in the Asian-Australian monsoon region (10°S–40°N, 40°–140°E). The purpose of this analysis is to further illustrate whether $R_v$ can well measure the similarity of two vector fields or not with observational data. The wind data used is NCEP-DOE reanalysis 2 data (Kanamitsu, et al., 2002). The climatological mean 850-hPa vector winds show a clear winter monsoon circulation characterized by northerly winds over the tropical and subtropical Asian regions in January and February (Figs. 3a, 3b). The spatial pattern of vector winds in January is very close to that in February, which corresponds to a very high $R_v$ (0.97). The spatial pattern of vector winds in January is less similar to that in April and October, which corresponds to a weak $R_v$ of 0.48 and -0.11, respectively. In August, the spatial pattern of 850-hPa winds is generally opposite to that in January, which corresponds to a negative $R_v$ (-0.64). The VSCs of 850-hPa vector winds between climatological January and each climatological month show a clear annual cycle characterized by a positive $R_v$ in the cold season (November-April) and a negative $R_v$ in the warm season (June-September) in the Asian-Australian monsoon region (Fig. 3f, solid line). Figure 3 illustrates that VSC can reasonably measure the pattern similarity of two vector fields. We also computed the VSCs of 850-hPa vector winds between climatological January and each individual month during the period from 1979 and 2005, respectively. The VSCs show a smaller spread in winter (January, February, and December) and summer (June, July, and August) months than during the transitional months such as April, May, and October (Fig. 3f). This indicates that the spatial patterns of vector winds have smaller inter-annual variation in summer and winter monsoon seasons than during the transitional seasons.

### 4 Construction of the VFE diagram

To measure the differences in two vector fields, a root-mean-square vector difference (RMSVD) is defined following Shukla and Saha (1974) with a minor modification:

$$\text{RMSVD} = \left[\frac{1}{N}\sum_{i=1}^{N}\left|\vec{A}_i - \vec{B}_i\right|^2\right]^{\frac{1}{2}}$$

where $\vec{A}_i$ and $\vec{B}_i$ are the original vectors. The RMSVD approaches zero when two vector fields become more alike in both vector length and direction. The square of RMSVD can be written as

$$\text{RMSVD}^2 = \frac{1}{N}\sum_{i=1}^{N}\left|\vec{A}_i - \vec{B}_i\right|^2$$

$$= \frac{1}{N}\sum_{i=1}^{N}\left(\left|\vec{A}_i\right|^2 + \left|\vec{B}_i\right|^2 - 2\left|\vec{A}_i \cdot \vec{B}_i\right|\right)$$

$$= \frac{1}{N}\sum_{i=1}^{N}|\vec{A}_i|^2 + \frac{1}{N}\sum_{i=1}^{N}|\vec{B}_i|^2 - \frac{2}{N}R_v \cdot \sqrt{\sum_{i=1}^{N}|\vec{A}_i|^2 \sum_{i=1}^{N}|\vec{B}_i|^2}$$

$$= \frac{1}{N}\sum_{i=1}^{N}|\vec{A}_i|^2 + \frac{1}{N}\sum_{i=1}^{N}|\vec{B}_i|^2 - 2R_v \cdot \sqrt{\frac{1}{N}\sum_{i=1}^{N}|\vec{A}_i|^2} \sqrt{\frac{1}{N}\sum_{i=1}^{N}|\vec{B}_i|^2}$$

With the support of equation (4), (5), (7), we obtain

$$RMSVD^2 = L_A^2 + L_B^2 - 2R_v \cdot L_A L_B \qquad (13)$$

The geometric relationship between RMSVD, $L_A$, $L_B$, and $R_v$ is shown in Figure 4, which is analogous to Figure 1 in Taylor (2001) but constructed by different quantities. It should be noted that RMSVD is computed from the two original sets of
vectors. However, the MSDNV in section 2 is computed using normalized vectors.

With the above definitions and relationships, we can construct a diagram that statistically quantifies how close two vector fields are to each other in terms of the $R_v$, $L_A$, $L_B$, and RMSVD. $L_A$ and $L_B$, measure the mean length of the vector fields $\vec{A}$ and $\vec{B}$, respectively. In contrast, RMSVD describes the magnitude of the overall difference between vector fields $\vec{A}$ and $\vec{B}$.
Vector field $\vec{B}$ can be called the "reference" field, usually representing some observed state. Vector field $\vec{A}$ can be regarded as a "test" field, typically a model-simulated field. The quantities in equation (13) are shown in Figure 5. The half circle represents the reference field, and the asterisk represents the test field. The radial distances from the origin to the points represents RMSL ($L_A$ and $L_B$), which is shown as dotted circles (Fig. 5). The azimuthal positions provide the vector similarity coefficient ($R_v$). The dashed lines measure the distance from the reference point, which represents the RMSVD.
Both the Taylor diagram and the VFE diagram are constructed based on the law of cosine. The differences between the two diagrams are summarized in Table 1. Indeed, the Taylor diagram can be regarded as a specific case of the VFE diagram, which is further interpreted in Appendix A.

## 5 Applications of the VFE diagram

### 5.1 Evaluating vector winds simulated by multiple models

A common application of the VFE diagram is to compare multi-model simulations against observations in terms of the patterns of vector winds. As an example, we assess the pattern statistics of climatological mean 850-hPa vector winds derived from the historical experiments by 19 CMIP5 models (Taylor et al., 2012) compared with the NCEP-DOE reanalysis 2 data during the period from 1979 to 2005. The evaluation was based on the monthly mean datasets from the first ensemble run of CMIP5 historical simulations and all datasets were regrided to a common grid of 2.5°×2.5°. Box averaging (bi-linear
interpolation) method was used to regrid the reanalysis data and model data to a coarse (finer) resolution. The RMSVD and

RMSL ($L_A$ and $L_B$) were normalized by the observed RMSL ($L_B$), i.e., RMSVD' = RMSVD/$L_B$, $L_A$' = $L_A$/$L_B$, and $L_B$' = 1. This leaves VSC unchanged and yields a normalized diagram as shown in Figure 6. The normalized diagram removes the units of variables and thus allows different variables to be shown in the same plot. The VSCs vary from 0.8 to 0.96 among 19 models, clearly indicating which model-simulated patterns of vector winds well resemble observations and which do not.

5   The diagram also clearly shows which models overestimate or underestimate the mean wind speed (Fig. 6). For example, in comparison with the reanalysis data, some models (e.g., 12, 19, 13, and 15) underestimate wind speed characterized by smaller normalized RMSLs over the Asian-Australian monsoon region in summer. In contrast, some models (e.g., 6 and 10) overestimate wind speed (Fig. 6a). In winter, most models overestimate the 850-hPa wind speed characterized greater normalized RMSLs (Fig. 6b).

To illustrate the performance of the VFE diagram in model evaluation, Figure 7 shows the spatial patterns of the climatological mean 850-hPa vector winds over the Asian-Australian monsoon region derived from the NCEP2 reanalysis and three climate models. Models 1 and 4 show a spatial pattern of vector winds very similar to the reanalysis data in summer, and $R_v$ reaches 0.96 and 0.95, respectively (Figs. 7a, 7c, 7e). In contrast, the spatial pattern of the vector winds

simulated by model 12 is less similar to the reanalysis data (Figs. 7a, 7g). For example, the reanalysis-based vector winds show stronger southwesterly winds over the southwestern Arabian Sea than the Bay of Bengal (Fig. 7a). However, an opposite spatial pattern is found in the same areas in model 12. More precisely, the southwesterly winds are weaker over the southwestern Arabian Sea than over the Bay of Bengal (Fig. 7g). $R_v$ reasonably gives expression to the lower similarity of the spatial pattern in the vector winds characterized by a smaller $R_v$ (0.86) in model 12 that is clearly lower than that (0.96)

in model 1. Figure 6 suggests that model 12 underestimates wind speed (normalized RMS wind speed is 0.78) in summer. In contrast, model 4 overestimates wind speed (normalized RMS wind speed is 1.35) in winter. These biases in wind speed can be identified in Figure 7. For example, model 12 generally underestimates the 850-hPa wind speed, especially over the Somali region in summer, compared with the reanalysis data (Figs. 7a, 7g). Model 4 overestimates the strength of easterly winds between 5°N and 20°N and westerly winds between the equator and 10°S in winter (Figs. 7b, 7f).

## 5.2 Other potential applications

Similar to the Taylor diagram (Taylor, 2001), the VFE diagram can be applied to the following aspects.

### 5.2.1 Tracking changes in model performance

To summarize the changes in the performance of a model, the points on the VFE diagram can be linked with arrows. For

example, similar to Figure 5 in Taylor (2011) the tails of the arrows represent the statistics for the older version, and the arrowheads point to the statistic for the newer version of the model. By doing so, the multiple statistical changes from the old version to the new version of the model can be clearly shown in the VFE diagram. The VFE diagram can also be combined

with the Taylor diagram to show the statistics for both scalar and vector variables in one diagram by plotting double coordinates because both diagrams are constructed based on the law of cosine.

### 5.2.2 Indicating the statistical significance of differences in model performance

As presented in Taylor (2001), one can qualitatively assess whether or not there are apparent differences in model performance by comparing ensemble simulations obtained from different models. The performance of two models can have a significant difference if the statistics from two groups of ensemble simulations are clearly separated from each other, and vice versa. As an illustration of this point, Figure 8 shows the normalized pattern statistics of the climatological mean 850-hPa vector winds derived from CMIP5 historical experiments during the period from 1979 to 2005. Models 12, 13, and 14 include 5, 9, and 9 ensemble runs, respectively. For a given model, all ensemble members of historical runs are forced in the same way, but each is initiated from a different point in the preindustrial control run (Taylor et al., 2012). Thus the differences between ensemble runs from the same model result from the sampling variability. In contrast, the differences between ensemble runs from different models are caused by both the sampling variability and model formulation differences. In figure 8, the symbols representing the same model show a close clustering, signifying that the sampling variability has less impact on the statistics of climatological mean vector winds. On the other hand, the symbols representing different models are clearly separated from each other. This suggests that the differences between models are much larger than the random sampling variability of individual models. Thus, the differences between models 12, 13, and 14 are likely to be significant. Models 12 and 13 are different versions of the same model. Compared with model 12, model 13 shows a similar RMSL but higher VSCs and smaller RMSVDs, which suggests that the improvement of model 13 beyond 12 is primarily due to the improvement of the spatial pattern of vector winds (Fig. 8). The ensemble member involved here is less than 10 and the statistics between models 12 and 13 are separated from each other by only a small distance on the VFE diagram, which may not be sufficient to conclude a significant difference between models 12 and 13. This is a shortcoming of this method, i.e. lacking quantitative evaluation on the significance of difference in model performance, and warrants further study. Specifically it may hard to determine the significance when the pattern statistics of two groups of simulations are not clearly separated from each other.

### 5.2.3 Evaluating model skill

The VFE diagram provides a concise evaluation of model performance. However it should be noted that, for a given VSC at relatively low value, the RMSVD does not decrease monotonically when the RMSL approaches the observed (Fig. 5). Thus a smaller RMSVD may not necessarily indicate a better model skill. To measure model skill in simulating vector fields, we developed two skill scores following the definition of model skill scores in Taylor (2001):

$$S_{v1} = \frac{4(1+R_v)}{(L_A/L_B+L_B/L_A)^2(1+R_0)} \qquad (14)$$

$$S_{v2} = \frac{4(1+R_v)^4}{(L_A/L_B+L_B/L_A)^2(1+R_0)^4} \qquad (15)$$

where $R_0$ is the maximum VSC attainable. $L_A$ and $L_B$ are the modeled and observed RMSL, respectively. $S_{v1}$ or $S_{v2}$ take values between zero (least skillful) and one (most skillful). Both skill scores can be shown as isolines in the VFE diagram, similar to Figure 10 and 11 in Taylor (2001). For a given $L_A/L_B$ the skill increases linearly with $R_v$. For a given $R_v$ the skill is proportional (inversely proportional) to $L_A$ when $L_A$ is smaller (greater) than $L_B$. Both skill scores, $S_{v1}$ and $S_{v2}$, take the VSC and the RMSL into account. However, $S_{v1}$ places more emphasis on the correct simulation of the vector length, whereas $S_{v2}$ pays more attention to the pattern similarity of the vector fields. Which statistical variable is more important depends on the application. For example, wind speed (measured by RMSL) may be the primary concern of model evaluation if one evaluates models for the purpose of wind power projection. In contrast, the pattern of vector winds (measured by VSC) may be the major concern if one evaluates model performance in simulating monsoon climate. The users should define or select appropriate skill scores based on their own applications because no skill score would be universally considered most appropriate.

## 6 The impact of observational uncertainty on model evaluation

It is known that observation data is uncertain due to many reasons, such as instrumental error, sampling error. Thus it is necessary to evaluate the impact of observational uncertainties on the result of model evaluation. Taylor (2001) presented a good approach to measure the observational uncertainty by showing the statistics of models relative to various observations on the Taylor diagram. Such an approach can also be applied here to assess the impact of observational uncertainty on the evaluation of simulated vector fields. For example, we can compute the normalized pattern statistics describing the climatological mean 850-hPa vector winds derived from CMIP5 models compared with six reanalysis datasets, respectively (Fig. 9). We assumed six reanalysis datasets, i.e. NECP/NCAR Reanalysis 1 (Kalnay et al., 1996), NCEP-DOE Reanalysis 2 (Kanamitsu, et al., 2002), ERA40 (Uppala, et al., 2005), ERA-interim (Dee, et al., 2011), JRA25 (Onogi, et al., 2007), and JRA55 (Kobayashi, et al., 2015; Harada, et al., 2016) are observational data here. The modeled patterns statistics against various reanalysis datasets are similar to each other, indicating that the observational uncertainty in vector winds has a minor impact on the evaluation of simulated climatological mean 850-hPa vector winds.

Note that the pattern statistics are less discriminable in figure 9 due to the overlapping of many symbols, although we use different symbols and colors to distinguish them from each other. To make the pattern statistics more clear, we propose an alternative way to show the observational uncertainty by comparing each model and observation with the mean of multiple observational estimates. If we assume various observational estimates are obtained independently and contain random noises those can contaminate the observational estimate. The random noises in various observational estimates could cancel out each other to a certain degree. Thus the mean of multiple observational estimates may be closer to the true value than the individual observational estimate. We therefore take the ensemble mean of six reanalysis datasets as a "reference data" and compute the pattern statistics of various models compared with the reference data to assess the model performance. Likewise, we can also measure the observational uncertainty by computing the pattern statistics of individual observational estimate

relative to the reference data. The pattern statistics derived from models and individual observations can be shown on the VFE diagram with different symbols (Fig. 10). By doing so, one can roughly estimate the impacts of observational uncertainty on the evaluation of model performance. For example, six reanalysis date sets show very close pattern statistics in summer characterized by high VSCs (0.986 – 0.994) and almost same RMSLs (0.986 – 1.021) as the reference data, which indicates a small observational uncertainty. Consequently, the observational uncertainty should have less impact on the evaluation of model performance. This is further supported by the comparison of figure 10 with figure 6. For example, the pattern statistics of CMIP5 models only show some minor changes when we replace the referenced NCEP-DOE reanalysis 2 datasets with the ensemble mean of six reanalysis (Figs. 6, 10).

## 7 Discussion and Conclusions

In this study, we devised a vector field evaluation (VFE) diagram based on the geometric relationship between three scalar variables, i.e., the vector similarity coefficient (VSC), root-mean-square length (RMSL), and root-mean-square vector difference (RMSVD). Three statistical variables in the VFE diagram are meaningful and easy to compute. VSC is defined by the arithmetic mean of the inner product of normalized vector pairs to measure the pattern similarity between two vector fields. Our results suggest that VSC can well describe the pattern similarity of two vector fields. RMSL measures the mean and variance of vector lengths (Equation A1). RMSVD measures the overall difference between two vector fields. The VFE diagram can clearly illustrate how much the overall RMSVD is attributed to the systematic difference in vector length versus how much is due to poor pattern similarity.

As discussed in Appendix A, three statistical variables can be computed with full vector fields (including both the mean and anomaly) or vector anomaly fields. One can compute three statistical variables using full vector fields if the statistics in both the mean state and anomaly need to be evaluated (Figs. 6, 8). Alternatively, one can compute three statistical variables using vector anomaly fields if the statistics in the anomaly are the primary concern. Under certain circumstance, e.g. the pattern of vector fields are highly homogeneous, the statistics of full vector fields could largely be dominated by the mean vector fields with a minor contribution from the anomaly fields (Equation A1-A4). Consequently, the statistics derived from different models may be very similar and difficult to separate from each other. In this case, one may want to assess the mean and anomaly fields, respectively. By doing so, the model performance in simulating vector anomaly fields can be better identified on the VFE diagram. The VFE diagram is devised to compare the statistics between two vector fields, e.g., vector winds usually comprise 2- or 3-dimensional vectors. One-dimensional vector fields can be regarded as scalar fields. In terms of the one-dimensional case, the VSC, RMSL, and RMSVD computed by anomaly fields become the correlation coefficient, standard deviation, and centered RMSE, respectively, and they are the statistical variables in the Taylor diagram. Thus, the Taylor diagram is a specific case of the VFE diagram. The Taylor diagram compares the statistics of scalar anomaly fields.

The VFE diagram is a generalized Taylor diagram that can compare the statistics of full vector fields or vector anomaly fields.

In practice, one may want to take latitudinal weight into account in the evaluation of spatial pattern of vector fields. This can be easily done by weighting the modeled and observed vector fields before computing VSC, RMSL, and RMSVD. Note that weighting should not be used during the computations of VSC, RMSL, and RMSVD to maintain their cosine relationship (Equation 13). The VFE diagram can also be easily applied to the evaluation of 3-dimensional vectors; however, we only considered 2-dimensional vectors in this paper. If the vertical scale of 3-dimensional vector variable is much smaller than its horizontal scale, e.g., vector winds, one may consider multiplying the vertical component by 50 or 100 to accentuate its importance. In addition, as with the Taylor diagram, the VFE diagram can also be applied to track changes in model performance, indicate the significance of the differences in model performance, and evaluate model skills. More applications of the VFE diagram could be developed based on different research aims in the future.

**Code availability**

The code used in the production of Figure 2 and 6a are available in the supplement to the article.

**Appendix A: The relationship between the VFE diagram and the Taylor diagram**

Consider two full vector fields $\vec{A}$ and $\vec{B}$:

$$\vec{A}_i = (x_{ai}, y_{ai}); \ i = 1, 2, \ldots, N$$

$$\vec{B}_i = (x_{bi}, y_{bi}); \ i = 1, 2, \ldots, N$$

$\vec{A}_i$ and $\vec{B}_i$ are 2-dimensional vectors. Each full vector field includes N vectors and can be broken into the mean and anomaly:

$$\vec{A}_i = \overline{\vec{A}} + \vec{A}'_i = (\overline{x}_a + x'_{ai}, \ \overline{y}_a + y'_{ai}); \ i = 1, 2, \ldots, N$$

$$\vec{B}_i = \overline{\vec{B}} + \vec{B}'_i = (\overline{x}_b + x'_{bi}, \ \overline{y}_b + y'_{bi}); \ i = 1, 2, \ldots, N$$

where $\overline{x}_a = \frac{1}{N}\sum_{i=1}^{N} x_{ai}$, $\overline{y}_a = \frac{1}{N}\sum_{i=1}^{N} y_{ai}$, $\overline{x}_b = \frac{1}{N}\sum_{i=1}^{N} x_{bi}$, $\overline{y}_b = \frac{1}{N}\sum_{i=1}^{N} y_{bi}$, $\overline{\vec{A}} = (\overline{x}_a, \overline{y}_a)$, $\overline{\vec{B}} = (\overline{x}_b, \overline{y}_b)$, $\vec{A}'_i = (x'_{ai}, y'_{ai})$,

$\vec{B}'_i = (x'_{bi}, y'_{bi})$

The standard deviation of the x- and y-component of vector $\vec{A}_i$ and $\vec{B}_i$ can be written as follows:

$$\sigma_{ax} = \sqrt{\frac{1}{N}\sum_{i=1}^{N}(x_{ai} - \overline{x}_a)^2} = \sqrt{\frac{1}{N}\sum_{i=1}^{N}{x'_{ai}}^2}, \ \sigma_{ay} = \sqrt{\frac{1}{N}\sum_{i=1}^{N}(y_{ai} - \overline{y}_a)^2} = \sqrt{\frac{1}{N}\sum_{i=1}^{N}{y'_{ai}}^2}$$

$$\sigma_{bx} = \sqrt{\frac{1}{N}\sum_{i=1}^{N}(x_{bi} - \overline{x}_b)^2} = \sqrt{\frac{1}{N}\sum_{i=1}^{N}{x'_{bi}}^2}, \ \sigma_{by} = \sqrt{\frac{1}{N}\sum_{i=1}^{N}(y_{bi} - \overline{y}_b)^2} = \sqrt{\frac{1}{N}\sum_{i=1}^{N}{y'_{bi}}^2}$$

The RMSL of vector field $\overline{\vec{A}}$ is written as follows:

$$L_A{}^2 = \frac{1}{N}\sum_{i=1}^{N}\left|\vec{A}_i\right|^2$$

$$= \frac{1}{N}\sum_{i=1}^{N}\left((\overline{x}_a + x'_{ai})^2 + (\overline{y}_a + y'_{ai})^2\right)$$

$$= \frac{1}{N}\sum_{i=1}^{N}(\overline{x}_a{}^2 + \overline{y}_a{}^2) + \frac{1}{N}\sum_{i=1}^{N}({x'_{ai}}^2 + {y'_{ai}}^2) + \frac{1}{N}\sum_{i=1}^{N}(2\overline{x}_a x'_{ai} + 2\overline{y}_a y'_{ai})$$

Given $\sum_{i=1}^{N} x'_{ai} = \sum_{i=1}^{N} y'_{ai} = 0$, $L_A{}^2$ can be written as:

$$L_A{}^2 = \frac{1}{N}\sum_{i=1}^{N}\left|\overline{\vec{A}}_i\right|^2 + \frac{1}{N}\sum_{i=1}^{N}\left|\vec{A}'_i\right|^2 \tag{A1}$$

$$= L_{\overline{A}}^2 + L_{A'}^2$$

where $L_{\overline{A}}^2 = \frac{1}{N}\sum_{i=1}^{N}\left|\overline{\vec{A}}\right|^2$, $L_{A'}^2 = \frac{1}{N}\sum_{i=1}^{N}\left|\vec{A}'_i\right|^2$

Similarly, we have

$$L_B{}^2 = L_{\overline{B}}^2 + L_{B'}^2 \tag{A2}$$

The VSC between vector fields $\vec{A}$ and $\vec{B}$:

$$R_v = \frac{1}{\sqrt{\sum_{i=1}^{N}\left|\vec{A}_i\right|^2}\sqrt{\sum_{i=1}^{N}\left|\vec{B}_i\right|^2}}\sum_{i=1}^{N}\vec{A}_i \cdot \vec{B}_i$$

$$= \frac{1}{NL_AL_B}\sum_{i=1}^{N}\left((\bar{x}_a + x'_{ai})(\bar{x}_b + x'_{bi}) + (\bar{y}_a + y'_{ai})(\bar{y}_b + y'_{bi})\right)$$

Given $\sum_{i=1}^{N} x'_{ai} = \sum_{i=1}^{N} y'_{ai} = 0$, we obtain

$$R_v = \frac{1}{NL_AL_B}\sum_{i=1}^{N}\left((\bar{x}_a\bar{x}_b + \bar{y}_a\bar{y}_b) + (x'_{ai}x'_{bi} + y'_{ai}y'_{bi})\right)$$

$$= \frac{1}{NL_AL_B}\left(\sum_{i=1}^{N}\bar{\vec{A}} \cdot \bar{\vec{B}} + \sum_{i=1}^{N}\vec{A}'_i \cdot \vec{B}'_i\right) \tag{A3}$$

$$= \frac{L_{\bar{A}}L_{\bar{B}}}{L_AL_B}R_{\bar{v}} + \frac{L_{A'}L_{B'}}{L_AL_B}R_{v'}$$

Where $R_{\bar{v}} = \frac{1}{NL_{\bar{A}}L_{\bar{B}}}\sum_{i=1}^{N}\bar{\vec{A}} \cdot \bar{\vec{B}} = \frac{\bar{\vec{A}}\cdot\bar{\vec{B}}}{L_{\bar{A}}L_{\bar{B}}} = \frac{\bar{\vec{A}}\cdot\bar{\vec{B}}}{\left|\bar{A}\right|\left|\bar{B}\right|}$ equals the cosine of included angle between two mean vectors. $R_{v'} = \frac{1}{NL_{A'}L_{B'}}\sum_{i=1}^{N}\vec{A}'_i \cdot \vec{B}'_i$ is the VSC between two vector anomaly fields.

5  The RMSVD$^2$ between vector fields $\vec{A}$ and $\vec{B}$:

$$\text{RMSVD}^2 = \frac{1}{N}\sum_{i=1}^{N}\left|\vec{A}_i - \vec{B}_i\right|^2$$

$$= \frac{1}{N}\sum_{i=1}^{N}\left((\bar{x}_a + x'_{ai} - \bar{x}_b - x'_{bi})^2 + (\bar{y}_a + y'_{ai} - \bar{y}_b - y'_{bi})^2\right)$$

$$= \frac{1}{N}\sum_{i=1}^{N}\left((\bar{x}_a - \bar{x}_b)^2 + (\bar{y}_a - \bar{y}_b)^2 + (x'_{ai} - x'_{bi})^2 + (y'_{ai} - y'_{bi})^2\right) \tag{A4}$$

$$= \frac{1}{N}\sum_{i=1}^{N}\left|\bar{\vec{A}} - \bar{\vec{B}}\right|^2 + \frac{1}{N}\sum_{i=1}^{N}\left|\vec{A}'_i - \vec{B}'_i\right|^2$$

Based on equation (A1), (A2), and (A4), we can conclude that the $L_A$, $L_B$, and RMSVD$^2$ derived from the full vector fields is equal to those derived from the mean vector fields plus those derived from the vector anomaly fields. The $R_v$ computed by two full vector fields is also determined by that derived from the mean state and anomaly (A3). This indicates that the VFE

diagram derived from the full vector fields takes the statistics in both the mean state and anomaly of the vector fields into account. The VFE diagram derived from the full vector fields is recommended for use if both the statistics in the mean state and anomaly are of great concern. On the other hand, the VFE diagram derived from vector anomaly fields can be used if the statistics in the anomaly are the primary concern. In this case, anomalous $L_A$, $L_B$, and $R_v$ and $RMSVD^2$ can be written, respectively, as follows:

$$L_{A'}^2 = \frac{1}{N}\sum_{i=1}^{N}\left|\vec{A}_i'\right|^2 = \frac{1}{N}\sum_{i=1}^{N}(x_{ai}'^2 + y_{ai}'^2) \tag{A5}$$

$$L_{B'}^2 = \frac{1}{N}\sum_{i=1}^{N}\left|\vec{B}_i'\right|^2 = \frac{1}{N}\sum_{i=1}^{N}(x_{bi}'^2 + y_{bi}'^2) \tag{A6}$$

$$R_{v'} = \frac{1}{\sqrt{\Sigma_{i=1}^N\left|\vec{A}_i'\right|^2}\sqrt{\Sigma_{i=1}^N\left|\vec{B}_i'\right|^2}}\sum_{i=1}^{N}\vec{A}_i'\cdot\vec{B}_i'$$

$$= \frac{1}{\sqrt{\Sigma_{i=1}^N(x_{ai}'^2 + y_{ai}'^2)}\sqrt{\Sigma_{i=1}^N(x_{bi}'^2 + y_{bi}'^2)}}\sum_{i=1}^{N}(x_{ai}'x_{bi}' + y_{ai}'y_{bi}') \tag{A7}$$

$$RMSVD_{v'}^2 = \frac{1}{N}\sum_{i=1}^{N}\left|\vec{A}_i' - \vec{B}_i'\right|^2$$

$$= \frac{1}{N}\sum_{i=1}^{N}((x_{ai}' - x_{bi}')^2 + (y_{ai}' - y_{bi}')^2) \tag{A8}$$

The vector fields $\vec{A}$ and $\vec{B}$ can be regarded as two scalar fields if we further assume that the y-component of both vector fields is equal to 0. Under this circumstance, equation (A5 – A8) can be written as follows:

$$L_{A'}^2 = \frac{1}{N}\sum_{i=1}^{N}x_{ai}'^2 = \sigma_{ax}^2$$

$$L_{B'}^2 = \frac{1}{N}\sum_{i=1}^{N}x_{bi}'^2 = \sigma_{bx}^2$$

$$R_{v'} = \frac{1}{\sqrt{\Sigma_{i=1}^N x_{ai}'^2}\sqrt{\Sigma_{i=1}^N x_{bi}'^2}}\sum_{i=1}^{N}x_{ai}'x_{bi}'$$

$$RMSVD_{v'}^2 = \frac{1}{N}\sum_{i=1}^{N}(x_{ai}' - x_{bi}')^2$$

$L_{A'}$ and $L_{B'}$ equal the standard deviation of the x-component of vector fields $\vec{A}$ and $\vec{B}$, respectively. $R_{V'}$ is the Pearson's correlation coefficient between the x-component of vector fields $\vec{A}$ and $\vec{B}$, and $RMSVD_{V'}^2$ is the centered RMS difference between the x-component of vector fields $\vec{A}$ and $\vec{B}$. The Taylor diagram is constructed using the standard deviation, correlation coefficient, and centered RMS difference (Taylor, 2001). Thus, the Taylor diagram can be regarded as a specific

case of the VFE diagram (i.e., for 1-dimensional anomaly fields). The VFE diagram is a generalized Taylor diagram which can be applied to multi-dimensional variables. In addition, the VFE diagram can evaluate model performance in terms of full vector fields or vector anomaly fields.

**Author contribution**

Z. Xu and Z. Hou are the co-first authors. Z. Xu constructed the diagram and led the study. Z. Hou and Z. Xu performed the
analysis. Z. Xu and Y. Han wrote the paper. All of the authors discussed the results and commented on the manuscript.

**Acknowledgements**

We acknowledge the World Climate Research Programme's Working Group on Coupled Modelling, which is responsible for CMIP, and we thank the climate modeling groups for producing and making their model output available. NCEP/NCAR Reanalysis 1 and NCEP-DOE Reanalysis 2 datasets were provided by the NOAA/OAR/ESRL PSD, Boulder, Colorado,
USA, through their website at http://www.esrl.noaa.gov/psd/. The ERA-40 and ERA-Interim Reanalysis datasets were provided by the European Centre for Medium-Range Weather Forecasts (ECMWF). The JRA25 and JRA55 Reanalysis datasets were provided from the Japanese 25- and 55-year Reanalysis projects carried out by the Japan Meteorological Agency (JMA).The study was supported jointly by the National Basic Research Program of China Project 2012CB956200, the National Key Technologies R&D Program of China (grant 2012BAC22B04), and the NSF of China Grant (41675105,
41475063).  This work was also supported by the Jiangsu Collaborative Innovation Center for Climate Change.

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

# Tables

Table 1 summarizing the difference between the Taylor diagram and the VFE diagram

|  | **Taylor diagram** | **VFE diagram** |
|---|---|---|
| **Purpose** | Evaluating scalar fields | Evaluating vector fields |
| **Composition** | Correlation coefficient (R), standard deviation (STD), centered RMSE | Vector similarity coefficient ($R_v$), RMS vector length (RMSL), RMSVD |
| **R vs $R_v$** | R: measuring the pattern similarity of scalar fields | $R_v$: measuring the pattern similarity of vector fields by considering vector length and direction simultaneously |
| **STD vs RMSL** | STD: measuring the variance of a scalar field | RMSL: measuring the mean and variance of vector lengths.. |
| **RMSE vs RMSVD** | centered RMSE: aggregating the magnitude of the errors between the simulated and observed anomaly fields | RMSVD: aggregating the magnitude of the overall difference between the simulated and observed vector fields. |

**Figures**

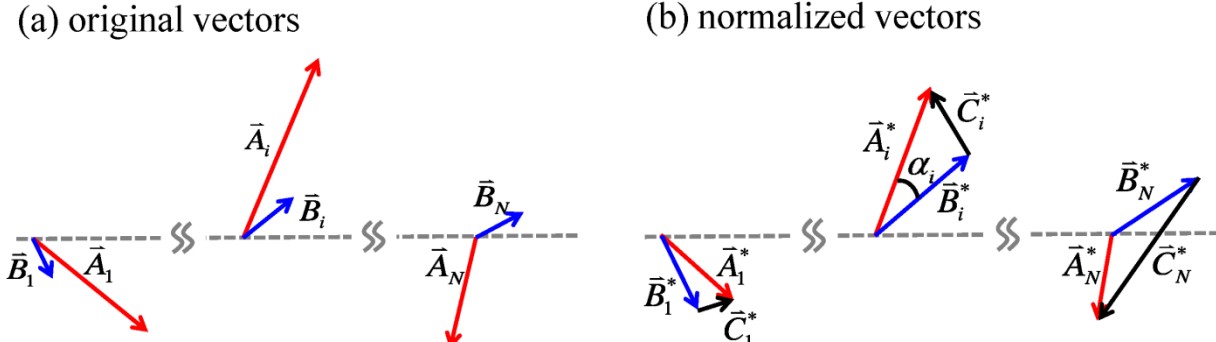

**Figure 1:** Schematic illustration of two vector sequences. (a) original vectors, (b) normalized vectors. The length of vector sequence $\vec{A}_i$ is systematically greater than that of vector sequence $\vec{B}_i$. The normalization only alters the lengths of vectors without changes in directions.

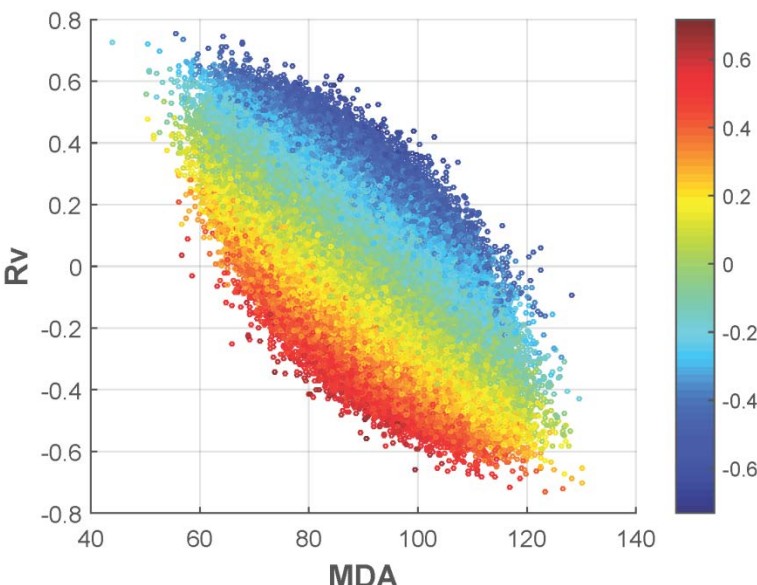

**Figure 2:** Scatter plot between the vector similarity coefficient ($R_v$) and mean difference of angle (MDA) derived from the reference vector field $\vec{A}$ and randomly generated vector field $\vec{B}$. There are $10^6$ random vector fields $\vec{B}$ included in the statistics. The colors denote the correlation coefficients between the vector length and the included angle between two vector sequences.

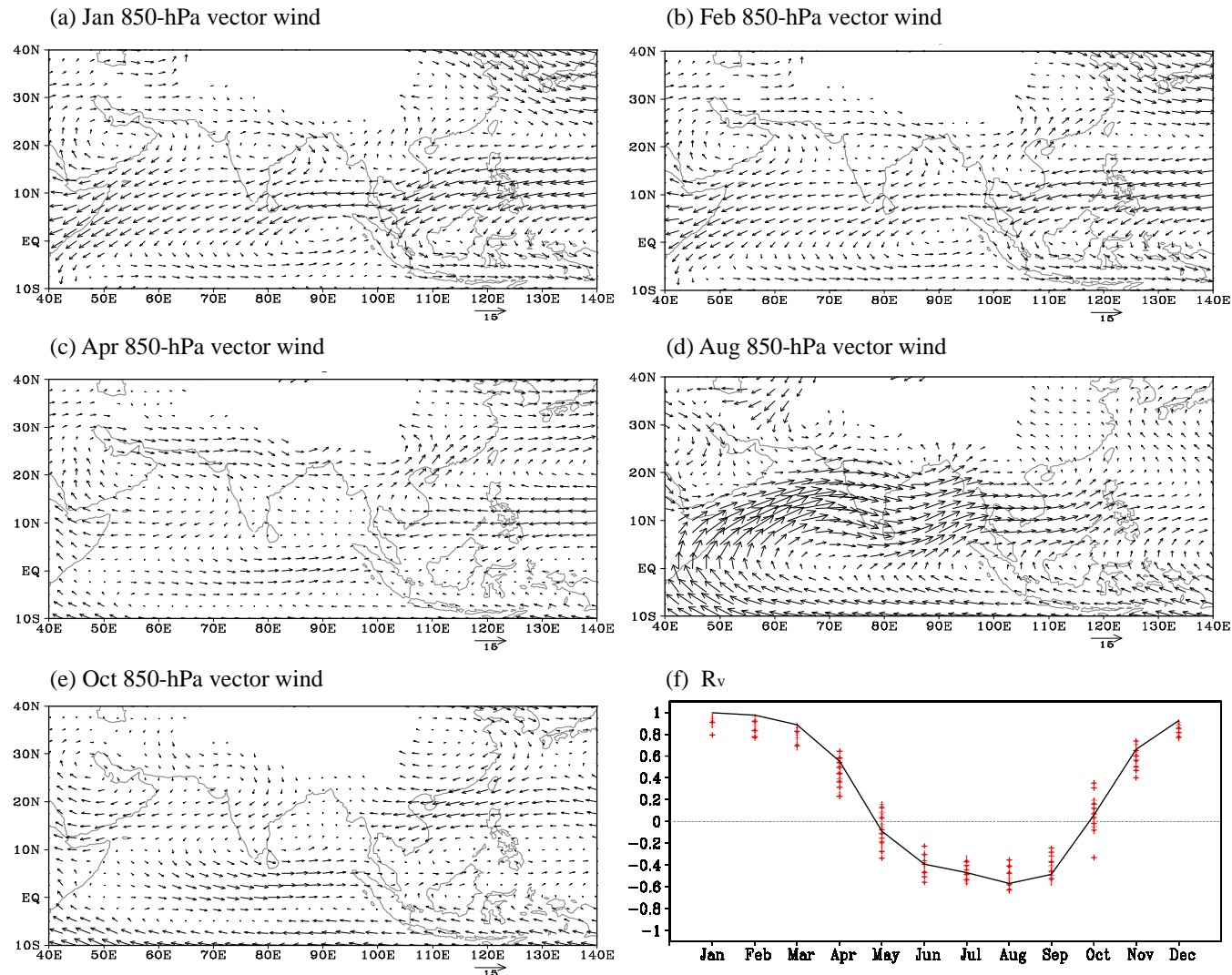

**Figure 3:** Climatological mean 850-hPa vector wind in (a) January, (b) February, (c) April, (d) August, and (e) October. (f) The vector similarity coefficients of 850-hPa vector winds between climatological mean January and 12 climatological months (Solid line). The "+" represents the VSC between the climatological mean vector winds in January and the vector winds in each individual month over the period of 1979–2014, respectively. There are 432 (12×36) "+" symbols. Monthly NCEP-NCAR reanalysis II data were used to produce this figure.

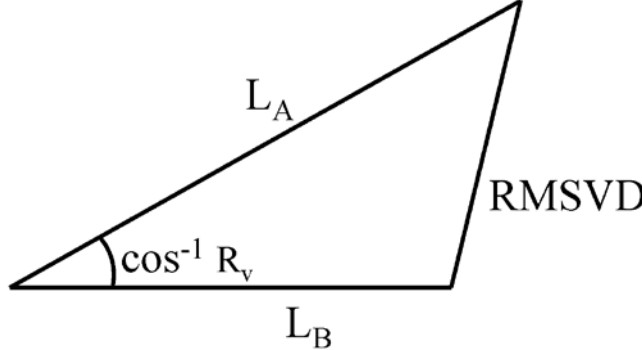

**Figure 4:** Geometric relationship among the vector similarity coefficient $R_v$, the RMS length $L_A$ and $L_B$, and RMS vector difference (RMSVD)

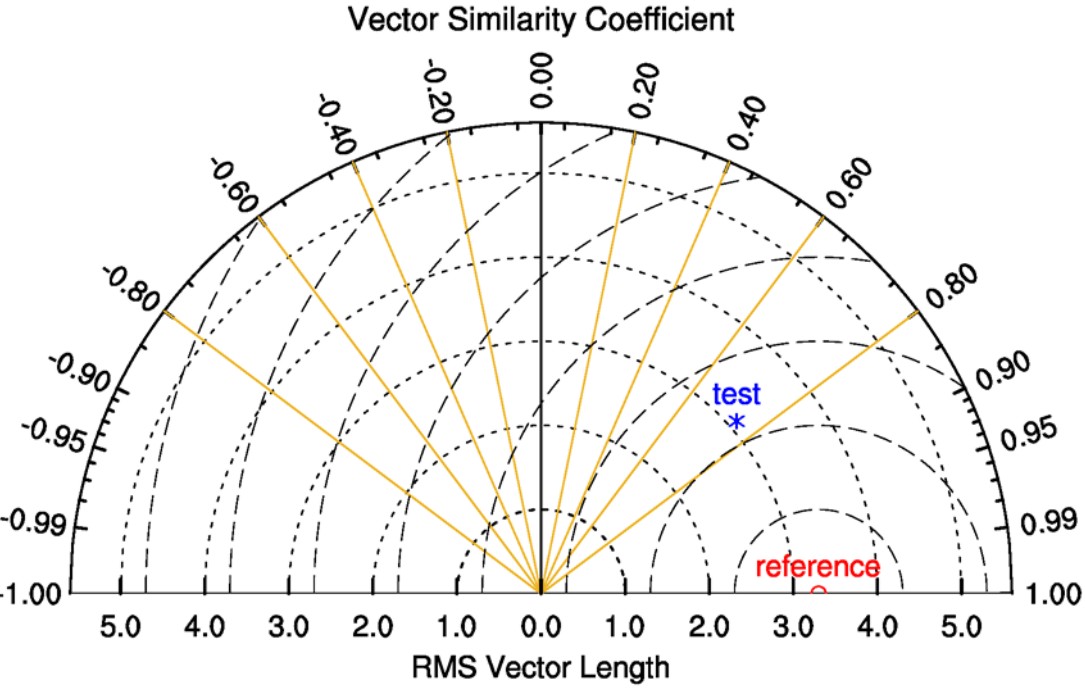

**Figure 5:** Diagram for displaying pattern statistics. The vector similarity coefficient between vector fields is given by the azimuthal position of the test field. The radial distance from the origin is proportional to the RMS length. The RMSVD between the test and reference field is proportional to their distance apart (dashed contours in the same units as the RMS length).

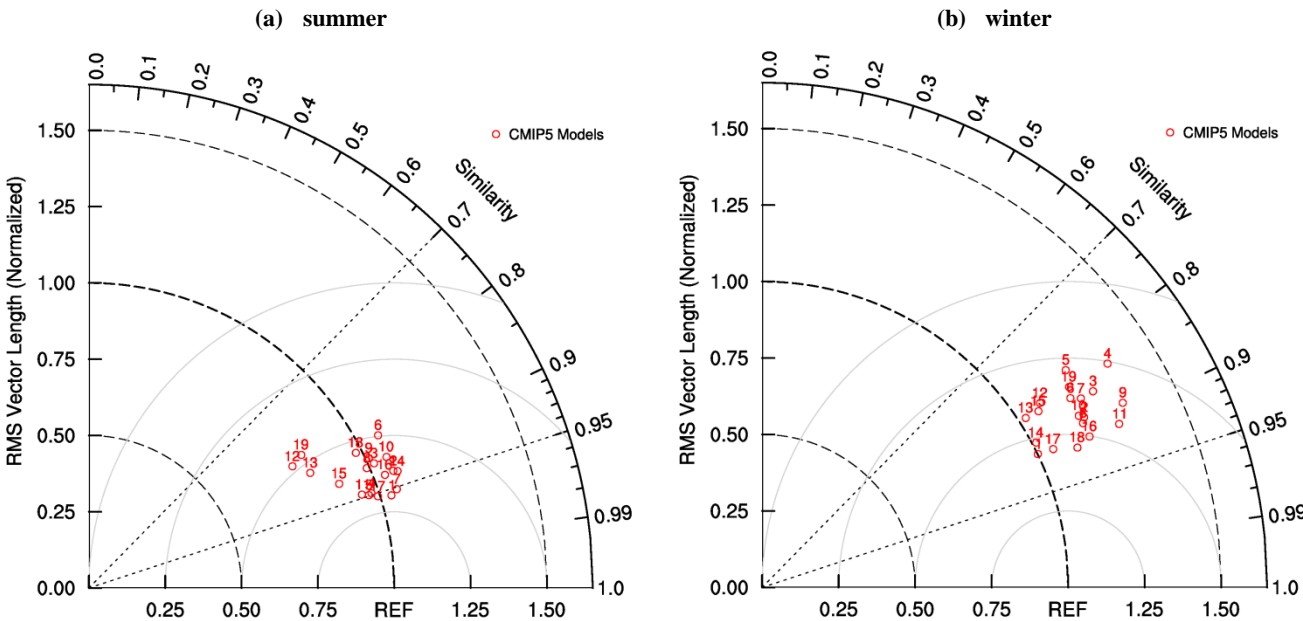

**Figure 6:** Normalized pattern statistics of climatological mean 850-hPa vector winds in the Asian-Australian monsoon region (10°S–40°N, 40°–140°E) in summer (June-July-August) and winter (December-January-February) among 19 CMIP5 models compared with the NCEP reanalysis 2 data during the period from 1979 to 2005. The RMS length and the RMSVD have been normalized by the RMS length derived from NCEP2. The data were excluded from the statistics in areas with a topography higher than 1500 m.

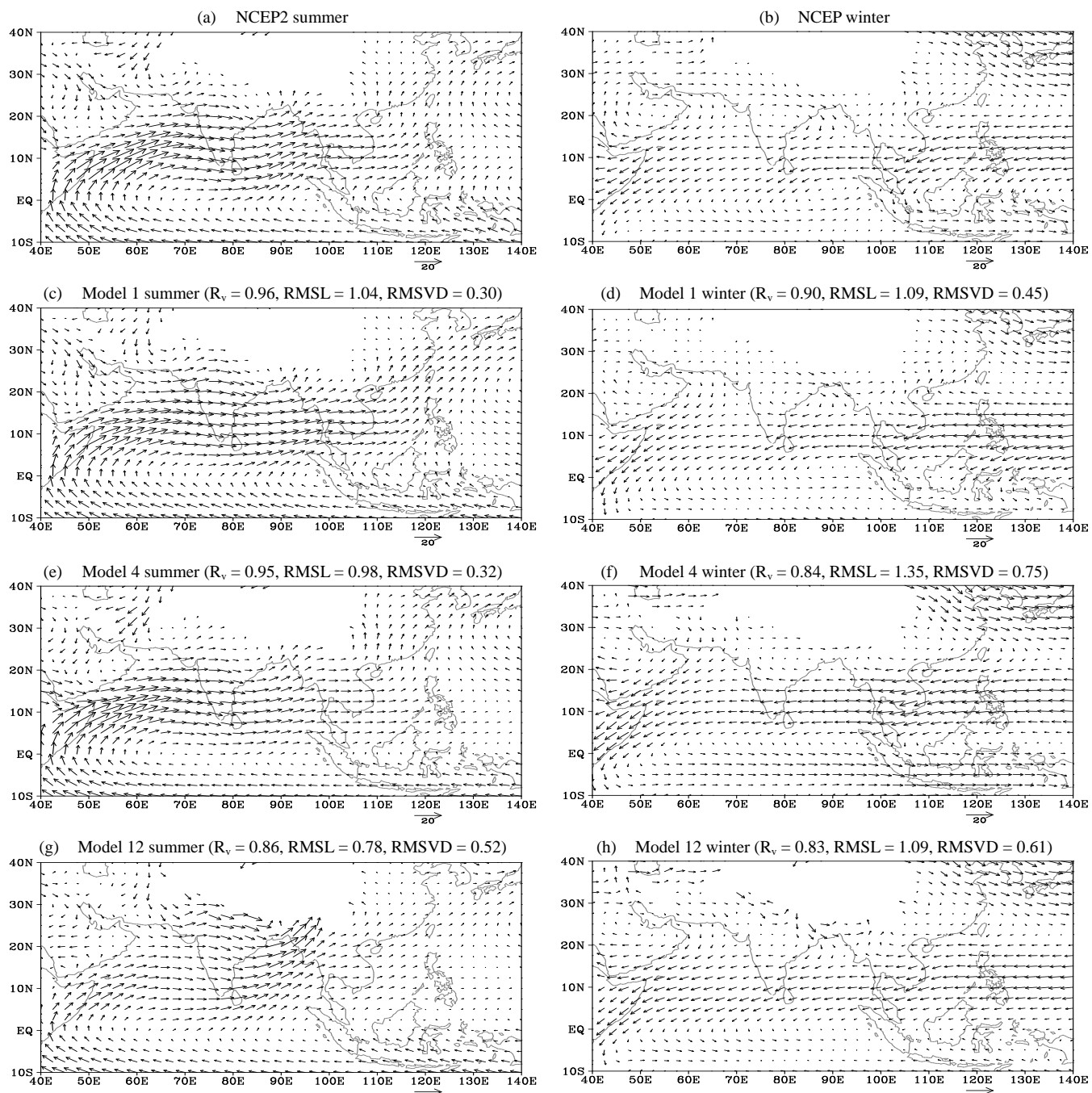

**Figure 7:** Climatological mean 850-hPa vector winds in summer and winter for the NCEP reanalysis II data and the results of historical simulations obtained from three CMIP5 models during the period 1979 to 2005. The vector similarity coefficient ($R_v$), normalized RMSL, and normalized RMSVD are also shown at the top of each panel. The vectors are set to a missing value in the areas with a topography higher than 1500 m.

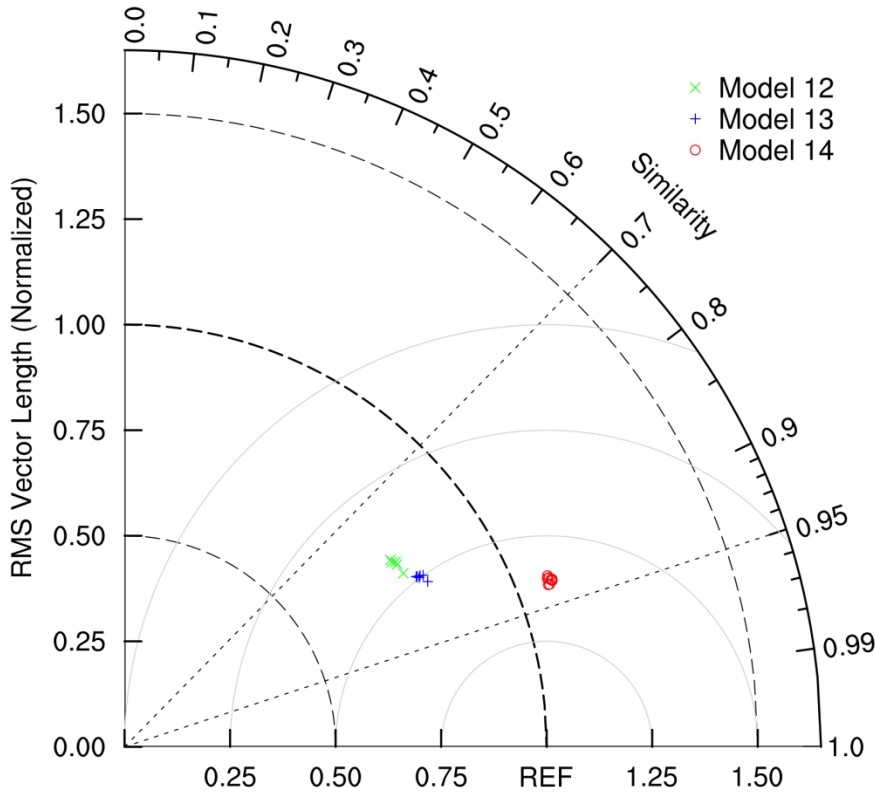

**Figure 8:** Normalized pattern statistics for climatological mean 850-hPa vector winds over the Asian-Austrian monsoon region (10°S–40°N, 40°–140°E) in summer (June-July-August) derived from each independent ensemble member by models 12, 13 and 14. The datasets used and regridding method are the same as those in Figure 6, except only three models are included here. Models 12, 13, and 14 include 5, 6, and 9 ensemble simulations which are obtained from CMIP5 historical experiments during the period from 1979 to 2005, respectively. The same type of symbols show a close clustering, and different types of symbols are clearly separate from each other, which suggests that the difference between different models are likely to be significant.

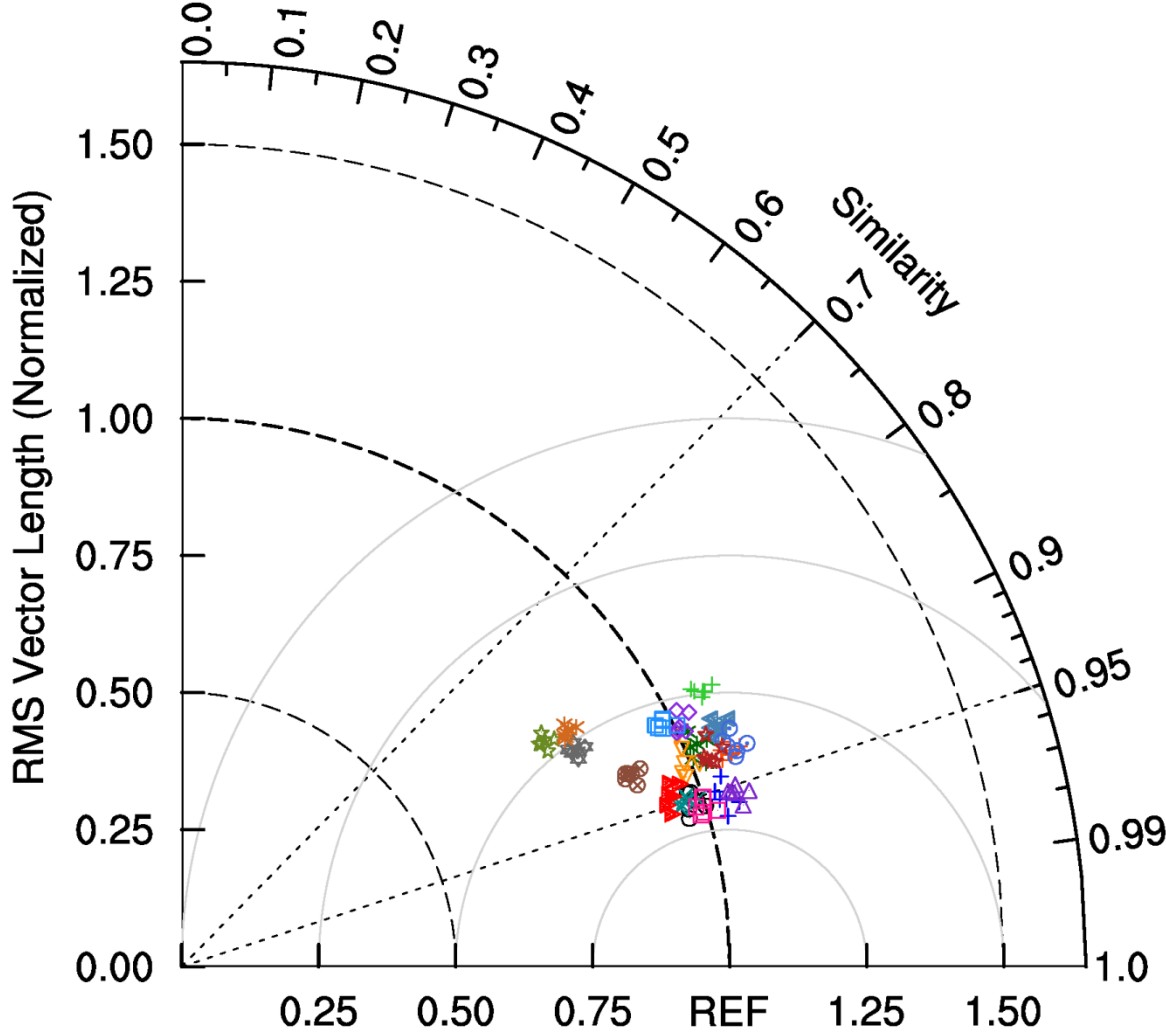

**Figure 9:** Normalized pattern statistics of climatological mean 850-hPa vector winds in the Asian-Australian monsoon region (10°S–40°N, 40°–140°E) among 19 CMIP5 models compared with the reanalysis data in summer (June-July-August). The climatological means were computed from the monthly data derived from CMIP5 historical simulations and reanalysis datasets during the period 1979–2005, except for the ERA-40 reanalysis with a time span of 1979–2002. Each CMIP5 model was compared with 6 reanalysis data, respectively. The symbols with same type of mark and color represent the statistics of an individual CMIP5 model compared with various reanalysis winds.

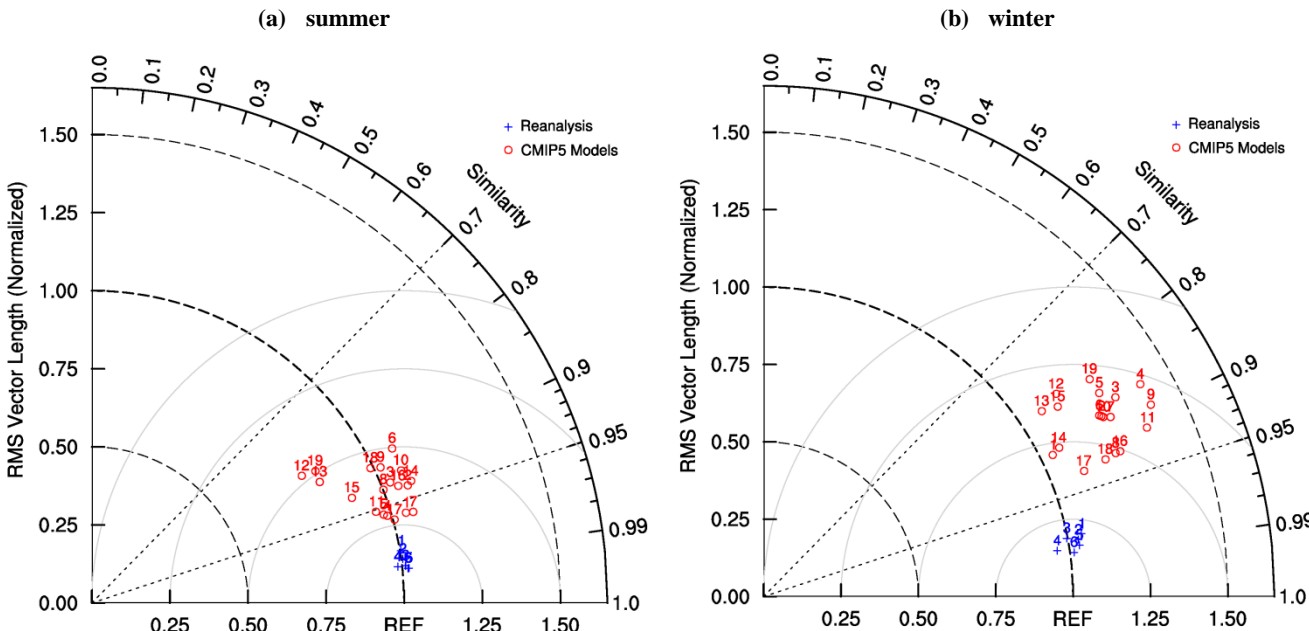

**Figure 10:** Normalized pattern statistics of climatological mean 850-hPa vector winds in the Asian-Australian monsoon region (10°S–40°N, 40°–140°E) derived from the historical simulations of 19 CMIP5 models compared with the multi-reanalysis mean during the period from 1979 to 2005. The RMSL and the RMSVD have been normalized by the RMSL derived from multi-reanalysis mean data. The data were excluded from the statistics in areas with topography higher than 1500 m. Six reanalysis data sets (NNRP, NCEP2, ERA40, ERA-Interim, JRA25, and JRA55) data were included in the statistics.