# Peer review of "A diagram for evaluating multiple aspects of model performance in simulating vector fields"

_Geoscientific Model Development, 2016_

## Referee Comment (RC1) · Anonymous Referee #1 · 5 Sep 2016

**A diagram for evaluating multiple aspects of model performance in simulating vector fields by Z. Xu, Z. Hou, Y. Han, and W. Guo**

The authors describe an extension of the Taylor diagram (Taylor, 2001) to compare 2-dimensional quantities such as horizontal wind fields from different sources in an analogous way to that described by Taylor (2001) for scalar quantities. This can be a useful extension to the widely Taylor diagrams when evaluating the output of climate models with observations or when comparing models again predecessor versions or other models.

I suggest major revisions to the manuscript before publication in Geoscientific Model Development addressing the points given below.

[Figure]

**General comments**

- A paragraph giving some guidance on the scientific interpretation of the proposed statistics would be helpful, possibly with examples demonstrating how "good" and "not so good" agreement between two data sets looks like. It would also be helpful to explicitly point to possible issues and limitations to keep in mind when looking at complex quantities such as the skill scores proposed by the authors (equations 13 and 14).

- An application of the proposed VFE diagram to evaluate the performance of models usually requires taking into account observational uncertainties as the reference data (at x=1, y=0; e.g., figure 9) are usually not the truth. This is particularly the case for quantities that have larger uncertainties than 850-hPa wind speed. I would like to see a discussion of possible issues and limitations as well as thoughts of how to deal with observational uncertainties in this type of diagram.

- What are the key messages of section 3.2 (relationship of VSC with MDA)? Things need to be put into context by providing a motivation for this analysis. The statements made have to be more precise, for example, "[...] a smaller MDA generally corresponds to a larger $R_v$, and vice versa." does not seem to provide a lot of useful information for the application and interpretation of the VFE diagram. Again, guidelines of how to interpret the proposed statistics in a scientifically meaningful way would be helpful.

- Section 5.1 gives an example for the application of the VFE diagram using 850-hPa wind speed from 19 CMIP5 models. There are, however, no details on the

model runs used (model experiment, ensemble members, etc.). It is also not clear to me what exactly you are comparing (multi-year annual means, monthly means, etc.). I presume the models have been regridded to a common grid? If so, which grid and which interpolation method has been applied? Also, I am missing a reference for the NCEP reanalysis data used. This section needs some rewriting to make clear what exactly has been done and what is being compared here. The current description is not sufficient to reproduce any of the results shown here, which is not acceptable for a scientific paper.

- Section 5.2.2 (statistical significance of differences): I am missing a clear definition of what the authors mean by "statistical significance of differences" in the context of 2-dim vector fields. The argument of "separated groups" without further explanations or governing equations is not precise enough for a scientific paper. I have the impression that the authors are rather speculating here than presenting any scientific evidence. This leads to contradictions within the section that need to be addressed. For example, the authors claim on p. 10, l. 27-28: "Thus, the differences between models 12, 13 [...] are likely to be significant.". The authors contradict this statement a few lines later (p. 11, l. 1-2): "[...] which may not be sufficient to conclude a significant difference between the three models, especially for models 12 and 13." leaving the reader confused. Also, it is once again not clear what data have been used and what quantities are being compared (models, model experiments, ensemble members, time period, averaging, time resolution, regridding, etc.).

**Specific comments**

- p. 5, l. 24, give equation numbers for RMSL
- p. 7, l. 8: replace "for a certain angle" with "by a certain angle"

- p. 7, l. 9: production → generation

- p. 7, l. 21: what do you mean by "the performance of R$_v$"?

- p. 9, l. 8: replace "a dotted contour" with, for instance, "dotted circles"

- p. 9, l. 9: "line" → "lines"

- p. 9, l. 15, insert "VFE" before "diagram"

- p. 11, l. 30: "anomalous scalar fields" → "scalar anomaly fields"

- p. 13, equation (A1): how did you get from line 2 to line 3? Shouldn't $(\bar{x}_{ai} + x\prime_{ai})^2 = \bar{x}_{ai}^2 + 2\bar{x}_{ai}x\prime_{ai} + x\prime_{ai}^2$? I.e., what happened to the term $2\bar{x}_{ai}x\prime_{ai}$?

- p. 14, equation (A4): similar to eqn. (A1), how did you get from line 2 to 3?

- p. 19: the added value of figure 2 seems rather limited, this figure could be deleted

- p. 20, "[...] and each randomly produced vector field" → "[...] and randomly generated vector fields"

- p. 20, delete "are" before "included in the statistics"

- p. 21, "vector similar coefficients" → "vector similarity coefficients"

- p. 21, "between January and 12 months (Solid line)": this formulation is not clear, please rephrase
* * *

---

## Referee Comment (RC2) · Anonymous Referee #2 · 17 Oct 2016

This paper introduces a counterpart to the widely used Taylor diagram that applies to vector fields. The authors define a vector similarity coefficient, Rv, and derive various properties of it. When paired with a measure of differences in vector lengths in two fields, one can construct a diagram with display features like those for scalar fields seen in Taylor diagrams. The authors provide examples that demonstrate the plausibility and utility of their construct. The diagnostic appears to be relatively straighforward to construct. Its interpretation mimics some of the properties of Taylor diagrams, which should render it accessible to viewers of the vector-based diagrams.

Other than a few comments below, the paper is well-written. I have not checked the details of the mathematics, but the mathematical development appears to be sound.

Minor comments:

[Figure]

Page 2, Line 25: The same shortcoming to be ascribed to the correlation of two scalar fields when a constant is added to one of them. However, in both the vector and scalar cases, the RMS difference would change, so the change would still be diagnosed.

P2, L 27-30: The same issue arises with Taylor diagrams for scalar fields – a change in a model might improve spatial correlation while increasing RMS differences. (However, the motivation to construct a vector-equvalent to the Taylor diagram still remains.)

P3m L24: The Rv definition is plausible as a measure of similarity, but it may help many readers if some motivation for it were given. For example, you are looking for a measure of similarity that recognizes how much the vectors point in the same direction; you want something that is independent of the average magnitude of the vectors; etc.

P7, L 17-18: The governing influence of longer vectors should be emphasized, for it suggests that error in determining the vector (say, an observational vector) does not undermine the Rv value when there are some vectors in the sequence that are relatively small in magnitude compared to the error, so long as the longer ones have small relative error.

P10, L10: Observations have error. If observations are used as the reference field, can the influence of that error on precise positioning of a model's mark on the diagram. If one assumes random error, can that be folded into the display on the diagram and thus illustrate when model results agree with observations to within the observational uncertainty? Or perhaps illustrate the relative size of some other "noise" quantities, such as the ranges of values due to interannual variability.

---

## Author Comment (AC2) · 28 Oct 2016

Thanks for the reviewer's valuable comments. Our point-by-point responses to reviewer's comments are listed below after each RC2 comment.

Page 2, Line 25: The same shortcoming to be ascribed to the correlation of two scalar fields when a constant is added to one of them. However, in both the vector and scalar cases, the RMS difference would change, so the change would still be diagnosed.

Response: Correlation coefficient is commonly used to measure the pattern similarity of two scalar fields. However computing correlation coefficient for the x- and y-component of two vector fields is not well suited for examining the pattern similarity of two vector fields as discussed in Page 2 Line 25. For example, If the x-component of vector field A adds a constant value, the correlation coefficients for both the x- and
y-component do not change, but the direction and length of vector A change, which suggests that the pattern of two vector fields are no longer identical. This is the reason why we develop a vector similarity coefficient in this paper. The centered RMSE used in Taylor diagram cannot detect the change of scalar field when a constant is added to it because the mean difference has been removed in the centered RMSE. As the reviewer argued, RMS difference can detect the change in mean. However RMS difference is not commonly used to measure the pattern similarity. For example, the pattern of scalar field does not change if the field adds a constant but RMS difference changes. Under such circumstance, the changes in RMS difference results from mean value change not the pattern change. Similarly, RMSVD is not suitable to measure pattern similarity, either. The VFE diagram developed in this paper can show how much RMSVD is attributed to the systematic difference in RMSL and how much is due to the poor pattern similarity (VSC).

P2, L27-30: The same issue arises with Taylor diagrams for scalar fields – a change in a model might improve spatial correlation while increasing RMS differences. (However, the motivation to construct a vector-equvalent to the Taylor diagram still remains.)

Response: It is true that a change in a model might improve spatial correlation while increase RMSE, because both the changes in correlation and standard deviation can affect RMSE. RMSE could increase if the modeled standard deviation increase compared with observation although correlation is improved. This is one of the important reasons why we need to examine multiple statistics rather than only one statistical variable. Taylor diagram provide a simple way to show multiple statistics on one diagram and can clearly show the how much change in centered RMSE can be attributed to the change in standard deviation and how much is due to the change in correlation coefficient. As the VFE diagram is a generalized Taylor diagram, VFE diagram can also provide similar information (Page 1 Line 21-23).

P3m L24: The Rv definition is plausible as a measure of similarity, but it may help many readers if some motivation for it were given. For example, you are looking for a

measure of similarity that recognizes how much the vectors point in the same direction; you want something that is independent of the average magnitude of the vectors; etc.

Response: We add one sentence to point out the motivation of developing Rv in section 2 in the revised manuscript. The sentence is "To measure the similarity between vector fields A âČŚ and B âČŚ, a vector similarity coefficient (VSC) should be able to recognize how much and to what degree the vectors are in the same direction and the vector lengths are proportional to each other. Thus VSC is defined as follows:".

P7, L 17-18: The governing influence of longer vectors should be emphasized, for it suggests that error in determining the vector (say, an observational vector) does not undermine the Rv value when there are some vectors in the sequence that are relatively small in magnitude compared to the error, so long as the longer ones have small relative error.

Response: We interpret the governing influence of longer vectors further in the revised manuscript to provide more insight of Rv. The sentences are rephrased as: "A positive (negative) Rv is observed when the 30 vector lengths and included angles are negatively (positively) correlated. This means that the patterns of two vector fields are closer (opposite) to each other when the included angles between the long vectors are small (large). Specifically, the rotation of shorter vector may not undermine RV too much as long as the longer vectors remain unchanged. In contrast, RV would be strongly undermined with the rotation of longer vectors. Simply put, the longer vectors generally play a more important role than the shorter vectors in determining Rv."

P10, L10: Observations have error. If observations are used as the reference field, can the influence of that error on precise positioning of a model's mark on the diagram. If one assumes random error, can that be folded into the display on the diagram and thus illustrate when model results agree with observations to within the observational uncertainty? Or perhaps illustrate the relative size of some other "noise" quantities, such as the ranges of values due to interannual variability.

Response: In the revised manuscript, we add a new section (Section 6) to address the observational uncertainty issue. The general idea is that we can use the mean of multiple observational estimates as reference data. All model results and individual observational estimate compare with the reference data and show these statistics on the VFE diagram. The observational uncertainty can be roughly estimated by the spread of symbols those describe the statistics of individual observational estimate against mean of multiple observational estimates. A new figure (Fig.10 in the revised manuscript) is also added to illustrate the observational uncertainty.

Please also note the supplement to this comment:
http://www.geosci-model-dev-discuss.net/gmd-2016-172/gmd-2016-172-AC2-supplement.pdf

**Supplement:**

[revised manuscript text omitted]
}} = \left(\overline{x}_a,\ \overline{y}_a\right)$, $\overline{\vec{B}} = \left(\overline{x}_b,\ \overline{y}_b\right)$, $\vec{A}_i' = \left(x_{ai}',\ y_{ai}'\right)$, $\vec{B}_i' = \left(x_{bi}',\ y_{bi}'\right)$

10  The standard deviation of the x- and y-component of vector $\vec{A}_i$ and $\vec{B}_i$ can be written as follows:

$\sigma_{ax} = \sqrt{\frac{1}{N}\sum_{i=1}^{N}(x_{ai} - \overline{x}_a)^2} = \sqrt{\frac{1}{N}\sum_{i=1}^{N} x_{ai}'^2}$, $\sigma_{ay} = \sqrt{\frac{1}{N}\sum_{i=1}^{N}\left(y_{ai} - \overline{y}_a\right)^2} = \sqrt{\frac{1}{N}\sum_{i=1}^{N} y_{ai}'^2}$

$\sigma_{bx} = \sqrt{\frac{1}{N}\sum_{i=1}^{N}(x_{bi} - \overline{x}_b)^2} = \sqrt{\frac{1}{N}\sum_{i=1}^{N} x_{bi}'^2}$, $\sigma_{by} = \sqrt{\frac{1}{N}\sum_{i=1}^{N}\left(y_{bi} - \overline{y}_b\right)^2} = \sqrt{\frac{1}{N}\sum_{i=1}^{N} y_{bi}'^2}$

The RMSL of vector field $\vec{A}$ is written as follows:

$$L_A^{\ 2} = \frac{1}{N}\sum_{i=1}^{N}\left|\vec{A}_i\right|^2$$

$$= \frac{1}{N}\sum_{i=1}^{N}\left((\overline{x}_a + x_{ai}')^2 + \left(\overline{y}_a + y_{ai}'\right)^2\right)$$

$$= \frac{1}{N}\sum_{i=1}^{N}\left(\overline{x}_a^{\ 2} + \overline{y}_a^{\ 2}\right) + \frac{1}{N}\sum_{i=1}^{N}\left(x_{ai}'^2 + y_{ai}'^2\right) + \frac{1}{N}\sum_{i=1}^{N}\left(2\overline{x}_a x_{ai}' + 2\overline{y}_a y_{ai}'\right)$$

Given $\sum_{i=1}^{N} x_{ai}' = \sum_{i=1}^{N} y_{ai}' = 0$, $L_A^{\ 2}$ can be written as:

$$L_A^{\ 2} = \frac{1}{N}\sum_{i=1}^{N}\left|\overline{\vec{A}}_i\right|^2 + \frac{1}{N}\sum_{i=1}^{N}\left|\vec{A}_i'\right|^2 \tag{A1}$$

$$= L_{\overline{A}}^2 + L_{A'}^2$$

15  where $L_{\overline{A}}^2 = \frac{1}{N}\sum_{i=1}^{N}\left|\overline{\vec{A}}\right|^2$, $L_{A'}^2 = \frac{1}{N}\sum_{i=1}^{N}\left|\vec{A}_i'\right|^2$

Similarly, we have

$$L_B^{\ 2} = L_{\overline{B}}^2 + L_{B'}^2 \tag{A2}$$

The VSC between vector fields $\vec{A}$ and $\vec{B}$:

$$R_v = \frac{1}{\sqrt{\sum_{i=1}^{N}\left|\vec{A}_i\right|^2}\sqrt{\sum_{i=1}^{N}\left|\vec{B}_i\right|^2}}\sum_{i=1}^{N}\vec{A}_i \cdot \vec{B}_i$$

$$= \frac{1}{NL_AL_B}\sum_{i=1}^{N}\left((\overline{x}_a + x'_{ai})(\overline{x}_b + x'_{bi}) + (\overline{y}_a + y'_{ai})(\overline{y}_b + y'_{bi})\right)$$

Given $\sum_{i=1}^{N}x'_{ai} = \sum_{i=1}^{N}y'_{ai} = 0$, we obtain

$$R_v = \frac{1}{NL_AL_B}\sum_{i=1}^{N}\left((\overline{x}_a\overline{x}_b + \overline{y}_a\overline{y}_b) + (x'_{ai}x'_{bi} + y'_{ai}y'_{bi})\right)$$

$$= \frac{1}{NL_AL_B}\left(\sum_{i=1}^{N}\overline{\vec{A}} \cdot \overline{\vec{B}} + \sum_{i=1}^{N}\vec{A}'_i \cdot \vec{B}'_i\right) \quad\quad\quad (A3)$$

$$= \frac{L_{\overline{A}}L_{\overline{B}}}{L_AL_B}R_{\overline{v}} + \frac{L_{A'}L_{B'}}{L_AL_B}R_{v'}$$

Where $R_{\overline{v}} = \frac{1}{NL_{\overline{A}}L_{\overline{B}}}\sum_{i=1}^{N}\overline{\vec{A}} \cdot \overline{\vec{B}} = \frac{\overline{\vec{A}}\cdot\overline{\vec{B}}}{L_{\overline{A}}L_{\overline{B}}} = \frac{\overline{\vec{A}}\cdot\overline{\vec{
[revised manuscript text omitted]

---

## Author Response (AR1)

Author's response to anomalous reviewer 1

Thanks for the reviewer's valuable comments. Our point-by-point responses to reviewer's comments are listed below.

5 • *A paragraph giving some guidance on the scientific interpretation of the proposed statistics would be helpful, possibly with examples demonstrating how "good" and "not so good" agreement between two data sets looks like. It would also be helpful to explicitly point to possible issues and limitations to keep in mind when looking at complex quantities such as the skill scores proposed by the authors (equations 13 and 14).*

The VFE diagram includes three statistical variables, i.e. the vector similarity coefficient (VSC), root-mean-
10 square length (RMSL), and root-mean-square vector difference (RMSVD). VSC was proposed in Section 2 of this study. A scientific interpretation of VSC was presented in Section 3 by discussing how the changes in vector length and angle will affect VSC. Three examples were also given in section 3.1, 3.2, and 3.3 to help readers to understand the meaning of VSC and its performance in describing the similarity of two vector fields. We have made some modifications to section 3 in the revised manuscript to improve the scientific interpretation of the
15 VSC. In terms of RMSL and RMSVD, we did not make further interpretation because they are very easy to understand.

One possible issue of the VFE diagram is pointed out in section 5.2.3 in the revised manuscript. Namely one should be cautious when using RMSVD measures model performance because a smaller RMSVD does not necessarily indicate a better model skill. Following Taylor (2001) we proposed two model skill scores to measure
20 model skill. Meanwhile, we also pointed out the caveat of the skill scores defined by equation 13 and 14. Namely, users should define or select appropriate skill scores based on their own applications because no skill score would be universally considered most appropriate.

• *An application of the proposed VFE diagram to evaluate the performance of models usually requires taking into*
25 *account observational uncertainties as the reference data (at x=1, y=0; e.g., figure 9) are usually not the truth. This is particularly the case for quantities that have larger uncertainties than 850-hPa wind speed. I would like to see a discussion of possible issues and limitations as well as thoughts of how to deal with observational uncertainties in this type of diagram.*

We add a new section (section 6) to address how to take the observational uncertainty into account in the model
30 evaluation. We present two approaches to show the observational uncertainty in the VFE diagram. One is Taylor's approach by showing the statistics of models relative to various observations in the VFE diagram. The

other approach is comparing models and individual observational estimates with the mean of multiple observational estimates. The spread of various observational estimates can be taken as an indicator of observational uncertainty. The observational uncertainty and model statistics can be shown on the VFE diagram.

*• What are the key messages of section 3.2 (relationship of VSC with MDA)? Things need to be put into context by providing a motivation for this analysis. The statements made have to be more precise, for example, "[...] a smaller MDA generally corresponds to a larger R$_v$, and vice versa." does not seem to provide a lot of useful information for the application and interpretation of the VFE diagram. Again, guidelines of how to interpret the proposed statistics in a scientifically meaningful way would be helpful.*

The section 3 presents the interpretation on VSC and includes three subsections. Given the fact that VSC is determined by both vector lengths and vector directions, we present how VSC is affected by MSD and MDA in section 3.1 and 3.2, respectively. As defined by equation (10) and (11), MSD measures the mean square difference of normalized vector lengths. MDA describes the mean difference of angles of paired vectors between two vector fields. The discussion in section 3.1 and 3.2 would help readers to understand the nature of VSC and whether or not VSC can reasonably reflect the similarity of two vector fields. Section 3 is properly revised to provide the motivation of the analysis and improve the scientific interpretation of the VSC. For example, we present the motivation of section 3.2 in the revised manuscript with the sentences "In previous section, the interpretation of VSC is based on the assumption that the paired vectors have a constant included angle. In this section, we will examine how VSC is affected by the difference of included angles in a more general case". We make more precise statement in the revised manuscript, e.g. "A smaller MDA indicates smaller differences in the directions of paired vectors and hence a higher similarity between the vector fields $\vec{A}$ and $\vec{B}$, suggesting that VSC can reasonably describe how close the vector directions between two vector fields are.". In addition the title of section 3.1 and 3.2 are changed to "Interpreting VSC based on its equation" and "Interpreting VSC based on random generated samples", respectively.

*• Section 5.1 gives an example for the application of the VFE diagram using 850-hPa wind speed from 19 CMIP5 models. There are, however, no details on the model runs used (model experiment, ensemble members, etc.). It is also not clear to me what exactly you are comparing (multi-year annual means, monthly means, etc.). I presume the models have been regridded to a common grid? If so, which grid and which interpolation method has been applied? Also, I am missing a reference for the NCEP reanalysis data used. This section needs some rewriting to*

*make clear what exactly has been done and what is being compared here. The current description is not sufficient to reproduce any of the results shown here, which is not acceptable for a scientific paper.*

In the revised manuscript, we present detailed information about the data used, regrinding method, and what we are comparing in the text and figure captions. For example, "As an example, we assess the pattern statistics of climatological mean 850-hPa vector winds derived from the historical experiments by 19 CMIP5 models (Taylor et al., 2012) compared with the NCEP-DOE reanalysis 2 data during the period from 1979 to 2005. The evaluation was based on the monthly mean datasets from the first ensemble run of CMIP5 historical simulations and all datasets were regrided to a common grid of 2.5°×2.5°. Box averaging (bi-linear interpolation) method was used to regrid the reanalysis data and model data to a coarse (finer) resolution.". References for the NCEP reanalysis datasets and other reanalysis datasets are also presented in the revised manuscript.

*• Section 5.2.2 (statistical significance of differences): I am missing a clear definition of what the authors mean by "statistical significance of differences" in the context of 2-dim vector fields. The argument of "separated groups" without further explanations or governing equations is not precise enough for a scientific paper. I have the impression that the authors are rather speculating here than presenting any scientific evidence. This leads to contradictions within the section that need to be addressed. For example, the authors claim on p. 10, l. 27-28: "Thus, the differences between models 12, 13 [...] are likely to be significant.". The authors contradict this statement a few lines later (p. 11, l. 1-2): "[...] which may not be sufficient to conclude a significant difference between the three models, especially for models 12 and 13." leaving the reader confused. Also, it is once again not clear what data have been used and what quantities are being compared (models, model experiments, ensemble members, time period, averaging, time resolution, regridding, etc.).*

The title of section 5.2.2 is reworded as "statistical significance of differences in model performance" to make it precise. We present more discussions on the statistical significance of differences in model performance in section 5.2.2. As discussed in Appendix A, the VFE diagram is a generalized Taylor diagram. Both are constructed based on the cosine law. Therefore the significance of model performance presented in this study can be understood in the same way as that in Taylor (2001, section 4.1). In terms of the "contradictions" of our discussion on the significance of model performance, we were trying to make an objective interpretation. The differences between models 12, 13, and 14 are "likely" to be significant because their statistics are clearly separated from each other. However, under certain circumstance, two groups of statistics are separated from each other may not be sufficient to conclude a significant difference, e.g., the sample size is small. This is similar to the significance test for scalar variables. The difference between two group data may not be statistically

significant even their mean value show a clear difference when the sample size is small. Currently the VFE diagram can only present a qualitatively evaluation on the statistical significance of differences in model performance. No quantitative evaluation is available yet. This is a drawback of our method and warrants for further study. We properly reworded our discussions in the revised manuscript. The data information is also

5  provided in the caption of figure 8.

*Specific comments*

• *p. 5, l. 24, give equation numbers for RMSL*

   Done

10  • *p. 7, l. 8: replace "for a certain angle" with "by a certain angle"*

   Done

• *p. 7, l. 9: production → generation*

   Done

• *p. 7, l. 21: what do you mean by "the performance of $R_v$"?*

15     The sentence has been reworded as "The purpose of this analysis is to further illustrate whether Rv can well measure the similarity of two vector fields or not with observational data."

• *p. 9, l. 8: replace "a dotted contour" with, for instance, "dotted circles"*

   Done. "a dotted contour" is replaced with "dotted circles"

• *p. 9, l. 9: "line" → "lines"*

20     Done

• *p. 9, l. 15, insert "VFE" before "diagram"*

   Done

• *p. 11, l. 30: "anomalous scalar fields" → "scalar anomaly fields"*

   Done

25  • *p. 13, equation (A1): how did you get from line 2 to line 3? Shouldn't $(\bar{x}_{ai} + x'_{ai})^2 = \bar{x}_{ai}^2 + 2\bar{x}_{ai}x'_{ai} + x'^2_{ai}$ ? I.e., what happened to the term $2\bar{x}_{ai}x'_{ai}$ ?*

   The term $\sum_{i=1}^{N} \bar{x}_{ai}x'_{ai}$  does no appear in equation (A1) because it equals 0. Equation (A1) is written as follow:

$$L_A{}^2 = \frac{1}{N}\sum_{i=1}^{N}\left|\vec{A}_i\right|^2$$

$$= \frac{1}{N} \sum_{i=1}^{N} \left( (\bar{x}_a + x'_{ai})^2 + (\bar{y}_a + y'_{ai})^2 \right)$$

$$= \frac{1}{N} \sum_{i=1}^{N} (\bar{x}_a^2 + \bar{y}_a^2) + \frac{1}{N} \sum_{i=1}^{N} (x'^2_{ai} + y'^2_{ai}) + \frac{1}{N} \sum_{i=1}^{N} (2\bar{x}_a x'_{ai} + 2\bar{y}_a y'_{ai})$$

The third term on the right-hand side of the equation can be written as:

$$\frac{1}{N} \sum_{i=1}^{N} (2\bar{x}_a x'_{ai} + 2\bar{y}_a y'_{ai}) = \frac{2\bar{x}_a}{N} \sum_{i=1}^{N} x'_{ai} + \frac{2\bar{y}_a}{N} \sum_{i=1}^{N} y'_{ai}$$

Given $\sum_{i=1}^{N} x'_{ai} = \sum_{i=1}^{N} y'_{ai} = 0$, we have $\frac{1}{N} \sum_{i=1}^{N} (2\bar{x}_a x'_{ai} + 2\bar{y}_a y'_{ai}) = 0$

We provide more details in the derivation of equation (A1) in the revised manuscript to make it easy to understand.

• *p. 14, equation (A4): similar to eqn. (A1), how did you get from line 2 to 3?*

For the same reason, the sum of anomaly equals to 0. Thus the terms $\sum_{i=1}^{N} (\bar{x}_{ai} x'_{ai})$, $\sum_{i=1}^{N} (\bar{x}_{ai} x'_{bi})$, are removed from the equation (A4).

• *p. 19: the added value of figure 2 seems rather limited, this figure could be deleted*

Figure 2 has been deleted in the revised manuscript.

• *p. 20, "[...] and each randomly produced vector field" →"[...] and randomly generated vector fields"*

Done

• *p. 20, delete "are" before "included in the statistics"*

Done

• *p. 21, "vector similar coefficients" ! "vector similarity coefficients"*

Done

• *p. 21, "between January and 12 months (Solid line)": this formulation is not clear, please rephrase*

The sentence has been reworded as "(f) The vector similarity coefficients of 850-hPa vector winds between climatological mean January and 12 climatological months (Solid line)" in the revised manuscript.

Author's response to anomalous reviewer 2

Thanks for the reviewer's valuable comments. Our point-by-point responses to reviewer's comments are listed below.

• *Page 2, Line 25: The same shortcoming to be ascribed to the correlation of two scalar fields when a constant is*
5   *added to one of them. However, in both the vector and scalar cases, the RMS difference would change, so the*
*change would still be diagnosed.*

   Correlation coefficient is commonly used to measure the pattern similarity of two scalar fields. However computing correlation coefficient for the x- and y-component of two vector fields is not well suited for examining the pattern similarity of two vector fields as discussed in Page 2 Line 25. For example, If the x-component of
10   vector field **A** adds a constant value, the correlation coefficients for both the x- and y-component do not change, but the direction and length of vector **A** change, which suggests that the pattern of two vector fields are no longer identical. This is the reason why we develop a vector similarity coefficient in this paper.

   The centered RMSE used in Taylor diagram cannot detect the change of scalar field when a constant is added to it because the mean difference has been removed in the centered RMSE. As the reviewer argued, RMS
15   difference can detect the change in mean. However RMS difference is not commonly used to measure the pattern similarity. For example, the pattern of scalar field does not change if the field adds a constant but RMS difference changes. Under such circumstance, the changes in RMS difference results from mean value change not the pattern change. Similarly, RMSVD is not suitable to measure pattern similarity, either. The VFE diagram developed in this paper can show how much RMSVD is attributed to the systematic difference in RMSL and how
20   much is due to the poor pattern similarity (VSC).

• *P2, L27-30: The same issue arises with Taylor diagrams for scalar fields – a change in a model might improve*
*spatial correlation while increasing RMS differences. (However, the motivation to construct a vector-equvalent*
*to the Taylor diagram still remains.)*
25   It is true that a change in a model might improve spatial correlation while increase RMSE, because both the changes in correlation and standard deviation can affect RMSE. RMSE could increase if the modeled standard deviation increase compared with observation although correlation is improved. This is one of the important reasons why we need to examine multiple statistics rather than only one statistical variable. Taylor diagram provide a simple way to show multiple statistics on one diagram and can clearly show the how much change in
30   centered RMSE can be attributed to the change in standard deviation and how much is due to the change in

correlation coefficient. As the VFE diagram is a generalized Taylor diagram, VFE diagram can also provide similar information (Page 1 Line 21-23).

*• P3m L24: The Rv definition is plausible as a measure of similarity, but it may help many readers if some motivation for it were given. For example, you are looking for a measure of similarity that recognizes how much the vectors point in the same direction; you want something that is independent of the average magnitude of the vectors; etc.*

We add one sentence to point out the motivation of developing Rv in section 2 in the revised manuscript. The sentence is "To measure the similarity between vector fields $\vec{A}$ and $\vec{B}$, a vector similarity coefficient (VSC) should be able to recognize how much and to what degree the vectors are in the same direction and the vector lengths are proportional to each other. Thus VSC is defined as follows:".

*• P7, L 17-18: The governing influence of longer vectors should be emphasized, for it suggests that error in determining the vector (say, an observational vector) does not undermine the Rv value when there are some vectors in the sequence that are relatively small in magnitude compared to the error, so long as the longer ones have small relative error.*

We interpret the governing influence of longer vectors further in the revised manuscript to provide more insight of $R_v$. The sentences are rephrased as: "A positive (negative) $R_v$ is observed when the 30 vector lengths and included angles are negatively (positively) correlated. This means that the patterns of two vector fields are closer (opposite) to each other when the included angles between the long vectors are small (large). Specifically, the rotation of shorter vector may not undermine $R_V$ too much as long as the longer vectors remain unchanged. In contrast, $R_V$ would be strongly undermined with the rotation of longer vectors. Simply put, the longer vectors generally play a more important role than the shorter vectors in determining $R_v$."

*• P10, L10: Observations have error. If observations are used as the reference field, can the influence of that error on precise positioning of a model's mark on the diagram. If one assumes random error, can that be folded into the display on the diagram and thus illustrate when model results agree with observations to within the observational uncertainty? Or perhaps illustrate the relative size of some other "noise" quantitites, such as the ranges of values due to interannual variability.*

In the revised manuscript, we add a new section (Section 6) to address the observational uncertainty issue. The general idea is that we can use the mean of multiple observational estimates as reference data. All model

results and individual observational estimate compare with the reference data and show these statistics on the VFE diagram. The observational uncertainty can be roughly estimated by the spread of symbols those describe the statistics of individual observational estimate against mean of multiple observational estimates. A new figure (Fig.10 in the revised manuscript) is also added to illustrate the observational uncertainty.

[revised manuscript text omitted]

addition, as with the Taylor diagram, the VFE diagram can also be applied to track changes in model performance, indicate the significance of the differences in model performance, and evaluate model skills. More applications of the VFE diagram could be developed based on different research aims in the future.

**Code availability**

5   The code used in the production of Figure 2 and 6a are available in the supplement to the article.

**Appendix A: The relationship between the VFE diagram and the Taylor diagram**

Consider two full vector fields $\vec{A}$ and $\vec{B}$:

$\vec{A}_i = (x_{ai}, y_{ai});\ \ i = 1, 2, \ldots, N$

$\vec{B}_i = (x_{bi}, y_{bi});\ \ i = 1, 2, \ldots, N$

5  $\vec{A}_i$ and $\vec{B}_i$ are 2-dimensional vectors. Each full vector field includes N vectors and can be broken into the mean and anomaly:

$\vec{A}_i = \overline{\vec{A}} + \vec{A}'_i = \left(\overline{x}_a + x'_{ai},\ \overline{y}_a + y'_{ai}\right);\ \ i = 1, 2, \ldots, N$

$\vec{B}_i = \overline{\vec{B}} + \vec{B}'_i = \left(\overline{x}_b + x'_{bi},\ \overline{y}_b + y'_{bi}\right);\ i = 1, 2, \ldots, N$

where $\overline{x}_a = \frac{1}{N}\sum_{i=1}^{N} x_{ai}$ , $\overline{y}_a = \frac{1}{N}\sum_{i=1}^{N} y_{ai}$ , $\overline{x}_b = \frac{1}{N}\sum_{i=1}^{N} x_{bi}$ , $\overline{y}_b = \frac{1}{N}\sum_{i=1}^{N} y_{bi}$ , $\overline{\vec{A}} = \left(\overline{x}_a,\ \overline{y}_a\right)$, $\overline{\vec{B}} = \left(\overline{x}_b,\ \overline{y}_b\right)$, $\vec{A}'_i = \left(x'_{ai},\ y'_{ai}\right)$,

$\vec{B}'_i = (x'_{bi},\ y'_{bi})$

10  The standard deviation of the x- and y-component of vector $\vec{A}_i$ and $\vec{B}_i$ can be written as follows:

$\sigma_{ax} = \sqrt{\frac{1}{N}\sum_{i=1}^{N}(x_{ai} - \overline{x}_a)^2} = \sqrt{\frac{1}{N}\sum_{i=1}^{N} x'^{\,2}_{ai}}$, $\sigma_{ay} = \sqrt{\frac{1}{N}\sum_{i=1}^{N}\left(y_{ai} - \overline{y}_a\right)^2} = \sqrt{\frac{1}{N}\sum_{i=1}^{N} y'^{\,2}_{ai}}$

$\sigma_{bx} = \sqrt{\frac{1}{N}\sum_{i=1}^{N}(x_{bi} - \overline{x}_b)^2} = \sqrt{\frac{1}{N}\sum_{i=1}^{N} x'^{\,2}_{bi}}$, $\sigma_{by} = \sqrt{\frac{1}{N}\sum_{i=1}^{N}\left(y_{bi} - \overline{y}_b\right)^2} = \sqrt{\frac{1}{N}\sum_{i=1}^{N} y'^{\,2}_{bi}}$

The RMSL of vector field $\vec{A}$ is written as follows:

$$L_A^{\ 2} = \frac{1}{N}\sum_{i=1}^{N} \left|\vec{A}_i\right|^2$$

$$= \frac{1}{N}\sum_{i=1}^{N} \left( (\overline{x}_a + x'_{ai})^2 + \left(\overline{y}_a + y'_{ai}\right)^2 \right)$$

$$= \frac{1}{N}\sum_{i=1}^{N} \left(\overline{x}_a^{\ 2} + \overline{y}_a^{\ 2}\right) + \frac{1}{N}\sum_{i=1}^{N} \left(x'^{\,2}_{ai} + y'^{\,2}_{ai}\right) + \frac{1}{N}\sum_{i=1}^{N} \left(2\overline{x}_a x'_{ai} + 2\overline{y}_a y'_{ai}\right)$$

Given $\sum_{i=1}^{N} x'_{ai} = \sum_{i=1}^{N} y'_{ai} = 0$, $L_A^{\ 2}$ can be written as:

$$L_A^{\ 2} = \frac{1}{N}\sum_{i=1}^{N} \left|\overline{\vec{A}}_i\right|^2 + \frac{1}{N}\sum_{i=1}^{N} \left|\vec{A}'_i\right|^2 \tag{A1}$$

$$= L_{\overline{A}}^2 + L_{A'}^2$$

15  where $L_{\overline{A}}^2 = \frac{1}{N}\sum_{i=1}^{N} \left|\overline{\vec{A}}\right|^2$ , $L_{A'}^2 = \frac{1}{N}\sum_{i=1}^{N} \left|\vec{A}'_i\right|^2$

Similarly, we have

$$L_B^{\ 2} = L_{\overline{B}}^2 + L_{B'}^2 \tag{A2}$$

The VSC between vector fields $\vec{A}$ and $\vec{B}$:

$$R_v = \frac{1}{\sqrt{\sum_{i=1}^{N}|\vec{A}_i|^2}\sqrt{\sum_{i=1}^{N}|\vec{B}_i|^2}}\sum_{i=1}^{N}\vec{A}_i \cdot \vec{B}_i$$

$$= \frac{1}{NL_A L_B}\sum_{i=1}^{N}\left((\bar{x}_a + x'_{ai})(\bar{x}_b + x'_{bi}) + (\bar{y}_a + y'_{ai})(\bar{y}_b + y'_{bi})\right)$$

Given $\sum_{i=1}^{N} x'_{ai} = \sum_{i=1}^{N} y'_{ai} = 0$, we obtain

$$R_v = \frac{1}{NL_A L_B}\sum_{i=1}^{N}\left((\bar{x}_a\bar{x}_b + \bar{y}_a\bar{y}_b) + (x'_{ai}x'_{bi} + y'_{ai}y'_{bi})\right)$$

$$= \frac{1}{NL_A L_B}\left(\sum_{i=1}^{N}\overline{\vec{A}}\cdot\overline{\vec{B}} + \sum_{i=1}^{N}\vec{A}'_i\cdot\vec{B}'_i\right) \qquad (A3)$$

$$= \frac{L_{\overline{A}}L_{\overline{B}}}{L_A L_B}R_{\bar{v}} + \frac{L_{A'}L_{B'}}{L_A L_B}R_{v'}$$

Where $R_{\bar{v}} = \frac{1}{NL_{\overline{A}}L_{\overline{B}}}\sum_{i=1}^{N}\overline{\vec{A}}\cdot\overline{\vec{B}} = \frac{\overline{\vec{A}}\cdot\overline{\vec{B}}}{L_{\overline{A}}L_{\overline{B}}} = \frac{\overline{\vec{A}}\cdot\overline{\vec{B}}}{|\overline{\vec{A}}||\overline{\vec{B}}|}$ 
[revised manuscript text omitted]

